# Immunotherapy and Vaccination in Surgically Resectable Non-Small Cell Lung Cancer (NSCLC)

**DOI:** 10.3390/vaccines9070689

**Published:** 2021-06-23

**Authors:** Li-Chung Chiu, Shu-Min Lin, Yu-Lun Lo, Scott Chih-Hsi Kuo, Cheng-Ta Yang, Ping-Chih Hsu

**Affiliations:** 1Division of Thoracic Medicine, Department of Internal Medicine, College of Medicine, Chang Gung Memorial Hospital at Linkou, Taoyuan City 33305, Taiwan; pomd54@cgmh.org.tw (L.-C.C.); smlin100@gmail.com (S.-M.L.); loyulun@hotmail.com (Y.-L.L.); chihhsikuo@gmail.com (S.C.-H.K.); yang1946@cgmh.org.tw (C.-T.Y.); 2Department of Thoracic Medicine, New Taipei Municipal Tu Cheng Hospital, New Taipei City 23652, Taiwan; 3Department of Medicine, College of Medicine, Chang Gung University, Taoyuan City 33302, Taiwan; 4Department of Internal Medicine, Taoyuan Chang Gung Memorial Hospital, Taoyuan City 33378, Taiwan; 5Department of Respiratory Therapy, College of Medicine, Chang Gung University, Taoyuan City 33302, Taiwan

**Keywords:** immunotherapy, programmed death-ligand 1 (PD-L1), cytotoxic T-lymphocyte-associated protein 4 (CTLA-4), immune checkpoint inhibitor, non-small cell lung cancer (NSCLC), cancer vaccination, early stage, surgery

## Abstract

Early-stage NSCLC (stages I and II, and some IIIA diseases) accounts for approximately 30% of non-small cell lung cancer (NSCLC) cases, with surgery being its main treatment modality. The risk of disease recurrence and cancer-related death, however, remains high among NSCLC patients after complete surgical resection. In previous studies on the long-term follow-up of post-operative NSCLC, the results showed that the five-year survival rate was about 65% for stage IB and about 35% for stage IIIA diseases. Platinum-based chemotherapy with or without radiation therapy has been used as a neoadjuvant therapy or post-operative adjuvant therapy in NSCLC, but the improvement of survival is limited. Immune checkpoint inhibitors (ICIs) have effectively improved the 5-year survival of advanced NSCLC patients. Cancer vaccination has also been explored and used in the prevention of cancer or reducing disease recurrence in resected NSCLC. Here, we review studies that have focused on the use of immunotherapies (i.e., ICIs and vaccination) in surgically resectable NSCLC. We present the results of completed clinical trials that have used ICIs as neoadjuvant therapies in pre-operative NSCLC. Ongoing clinical trials investigating ICIs as neoadjuvant and adjuvant therapies are also summarized.

## 1. Introduction

The global incidence of lung cancer has prominently increased among the various cancers in the last three decades. Lung cancer has become the leading cause of cancer-related deaths in both males and females [1,2]. It is histologically classified as non-small cell lung cancer (NSCLC) and small cell lung cancer (SCLC); NSCLC accounts for 85% of cases [3,4]. Surgery remains the main treatment for early-stage NSCLC (stages I and II, and some IIIA diseases), and approximately 30% of NSCLC patients present with the surgically resectable disease at initial diagnosis [5]; however, the risk of disease recurrence and cancer-related mortality are high, even for those NSCLC patients receiving complete resection [5,6]. Previous studies focusing on the long-term follow-up of post-operative NSCLC have shown that the five-year survival rate is lower than 70% for IB and about 35% for IIIA diseases [6,7]. Platinum-based chemotherapy has been recommended as a post-operative adjuvant therapy for stages II to IIIA patients in the past 20 years [6,7]. Post-operative adjuvant chemotherapy decreases the disease recurrence rate by about 15% and the mortality rate at five years by about 5% [6,7]. Platinum-based chemotherapy, with or without radiation therapy, has been used as induction neoadjuvant therapy before surgery; however, the improvement of survival in resectable NSCLC patients is still limited [6,7]. A recent pivotal clinical study (ADAURA) showed that the third-generation epidermal growth factor receptor (EGFR)-tyrosine kinase inhibitor (TKI) osimertinib significantly reduced the disease recurrence rate in stage IB to IIIA resected EGFR-mutated NSCLC patients [8]; however, the advances in neoadjuvant and post-operative adjuvant therapies for surgically resectable NSCLC have been very limited over the last three decades.

Immunotherapies are a new therapeutic modality, which has been studied and used for the treatment of advanced NSCLC in the past 10 years [9,10]; for example, anti-programmed cell death protein-1 (PD-1)/programmed death-ligand 1 (PD-L1) immune checkpoint inhibitors (ICIs) have been developed and widely studied in clinical trials, and have been used to treat advanced NSCLC. The clinical trials showed that immunotherapies targeting the PD-1/PD-L1 axis have a promising response (~45%) and can significantly prolong the survival of metastatic NSCLC patients [9,10]. Therefore, the application of immunotherapy in early-stage NSCLC has been explored in recent studies [11].

Here, we review the immunotherapies that have been studied and used in the treatment of NSCLC patients. We also review clinical studies using immunotherapies in resectable early-stage NSCLC and that have sought better strategies for applying immunotherapies in patients. 

## 2. Current Immune Checkpoint Inhibitors in Advanced Non-Small Cell Lung Cancer (NSCLC)

### 2.1. Anti-Programmed Cell Death Protein-1 (PD-1)/Programmed Death-Ligand 1 (PD-L1) Immune Checkpoint Antibodies in NSCLC

The immune checkpoint receptor PD-1 (known as CD279), which is located on the surface of immune cells (i.e., T-cells, B-cells, and myeloid cells), is engaged in immune regulation [12,13,14]. PD-L1 (known as B7-H1 or CD274) and PD-L2 (known as B7-DC or CD273) are the ligands of the PD-1 receptor, which interacts with these ligands (PD-L1 and PD-L2) to deliver a negative signal to the human immune system. PD-L1 is widely expressed in normal tissue cells of human organs, including the vascular endothelium, pancreas, brain, and cornea. In healthy human tissue cells, the binding of PD-1 and PD-L1 downregulates the survival and effector function of CD8+ T-cells, to induce T-cell tolerance and escape from host immunity. Previous studies have shown that dysregulation and deficiency of the PD-1/PD-L1 axis can cause human autoimmune diseases [12,13,14,15]; for example, the PD-1/PD-L1 axis plays an important role in the maintenance of immune tolerance in pancreatic islet cells for the prevention of type 1 diabetes mellitus [16]. PD-L1 expression has been found in various cancers, and cancer cells can use the PD-1/PD-L1 immune checkpoint to prevent being killed by host antitumor immune responses [17,18,19]. Therefore, anti-PD-1/PDL-1 immune checkpoint inhibitors (ICIs) have been developed and investigated as anti-cancer therapy agents in various cancers, including melanoma, lymphoma, hepatocellular carcinoma, colorectal cancer, head, and neck squamous cell carcinoma, and NSCLC [17,18,19]. 

Anti-PD-1/PDL-1 ICIs have been used worldwide as a front-line therapy for the treatment of advanced NSCLC [9,20,21]. Currently, there exist five anti-PD-1/PD-L1 ICIs—pembrolizumab, nivolumab, cemiplimab (anti-PD-1), atezolizumab, and durvalumab (anti-PD-L1)—which have been proven to be effective in treating advanced NSCLC, based on the results of several large and pivotal clinical trials [20,21,22,23,24,25,26,27,28,29,30,31,32,33,34]. Therefore, these five anti-PD-1/PD-L1 ICIs have been approved by the U.S. Food and Drug Administration (FDA), for use as a first-line therapy for advanced NSCLC [20,21,22,23,24,25,26,27,28,29,30,31,32,33,34]. The tumor surface PD-L1 expression level of NSCLC is a favorable predictive factor associated with the treatment response of anti-PD-1/PD-L1 ICIs [35,36,37]. In NSCLC patients with strong PD-L1 expression and tumor proportion scores (TPS) ≥50%, single ICI therapy with pembrolizumab, atezolizumab, or cemiplimab has shown good treatment response rates (of 35–45%) in previous clinical trials (KEYNOTE-024, IMpower110, and EMPOWER-Lung 1) [22,28,32]. In these three clinical trials, NSCLC patients with strong tumor surface PD-L1 expression (TPS ≧ 50%) who received first-line single ICI therapy (pembrolizumab, atezolizumab, or cemiplimab) had significantly longer survival than control patients receiving conventional chemotherapy [22,28,32]. 

The addition of anti-PD-1/PD-L1 ICIs to other treatment modalities, including chemotherapy, angiogenesis inhibitors, and radiation therapy, can synergize to improve the treatment efficacy for NSCLC, despite a high tumor surface PD-L1 expression level [23,24,25,26,27,28,29,30,31,32,33,34]. In the KEYNOTE-189 and KEYNOTE-407 trials, pembrolizumab combined with platinum-based chemotherapy significantly improved the overall survival, when compared with platinum-based chemotherapy alone, in both squamous and non-squamous NSCLC [24,25]. Durvalumab (anti-PD-L1), used as consolidation therapy in post-chemoradiotherapy and unresectable stage III NSCLC, was explored in the PACIFIC clinical trial, which led to significantly longer progression-free survival (PFS) and overall survival than in the placebo group [33,34]. Therefore, durvalumab is the first and only anti-PD-1/PD-L1 ICI authorized for consolidation therapy in post-chemoradiotherapy and unresectable stage III NSCLC patients. There exist some unfavorable genomic alternations which affect the efficacy of anti-PD-1/PD-L1 ICIs in NSCLC. 

A majority of NSCLC, especially adenocarcinomas harbor driver mutations and can be classified as oncogene-addicted NSCLC. Most oncogene-addicted NSCLC had effective target therapies to their driver mutations including EGFR, anaplastic lymphoma kinase (ALK), ROS1, BRAF, MET, HER2, RET, K-RAS, and NTRK [38,39]. Several previous clinical studies have shown that anti-PD-1/PD-L1 ICIs had a significantly lower response rate and shorter survival in NSCLC patients harboring EGFR, ALK or ROS-1 mutations than non-oncogene-addicted NSCLC patients [40,41,42]; therefore, EGFR-, ALK-, or ROS1- mutated NSCLC patients are generally not recruited in most clinical trials investigating the efficacy of first-line anti-PD-1/PD-L1 ICIs [22,23,24,25,26,27,28]. To date, several TKIs targeting EGFR, ALK, and ROS1 mutations had shown promising efficacy in treating EGFR-,ALK-,or ROS1- mutated NSCLC patients (60–80% response rate and 10–30 months of PFS). EGFR-TKIs (e.g., gefitinib, erlotnib, afatinib, dacomitinib, and osimertinib), ALK inhibitors (ex. crizotinib, ceritinib, alectinib, brigatinib, and lorlatinib), and ROS1 inhibitor (ex. crizotinib) had been approved and wildly used in the treatment of EGFR-,ALK-,or ROS1- mutated NSCLC patients [38,40,41,42]. Previous studies reported that the efficacy of anti-PD-L1 ICIs in NSCLC patients with BRAF, HER2, MET, KRAS, or RET mutations were close to unselected NSCLC patients [43,44]. For NSCLC with rare driver mutations such as BRAF, HER2, MET, KRAS, or RET, ICIs are treatment choices for these patients before reliable target therapies available. 

IMpower 150 is one anti-PD-1/PD-L1 ICI clinical trial that included EGFR- and ALK- mutated NSCLC patients in the study [30]. In the IMpower 150 trial, the combination therapy of bevacizumab + carboplatin + paclitaxel + atezolizumab significantly benefited NSCLC patients with the EGFR or ALK mutation in both PFS and OS. Previous studies have demonstrated that the addition of an anti-angiogenesis agent, such as bevacizumab (anti-vascular endothelial growth factor), could increase the efficacy of EGFR-TKIs and cytotoxic chemotherapy in advanced EGFR-mutated NSCLC [45,46]. Together, these results suggest that the addition of anti-angiogenic agents is needed when anti-PD-1/PD-L1 ICIs are used in NSCLC patients with EGFR and ALK mutations. More studies may be needed to explore the use and efficacy of anti-angiogenesis agents in combination with anti-PD-1/PD-L1 ICIs in NSCLC. 

Avelumab is an anti-PD-L1 ICI that has not been approved by U.S. FDA to be used in NSCLC, as it failed to show a survival benefit, when compared with chemotherapy in platinum-treated advanced NSCLC patients, in a previous clinical trial (JAVELIN Lung 200) [47]. Another clinical trial JAVELIN Lung 100 is currently ongoing, to investigate the use of avelumab in advanced NSCLC [48].

### 2.2. Anti-Cytotoxic T-Lymphocyte-Associated Protein 4 (CTLA-4) Antibodies

Cytotoxic T-lymphocyte-associated protein 4 (CTLA-4, also known as CD152) and CD28 are protein receptors that have similar structures and are widely expressed on T-cells. CTLA-4 and CD28 share the same ligands, CD80 (also known as B7.1) and CD86 (also known as B7.2), on antigen-presenting cells (APCs) [17,49,50]. When a T-cell receptor (TCR) is engaged by an antigen peptide to induce antigen recognition, the co-stimulation of CD28 amplifies TCR signaling, to promote T-cell activation, proliferation, differentiation, and cytokine production [17,49,50]. CTLA-4 is usually located intracellularly in resting T-cells and translocates to the cell surface when CD28 binds to the co-stimulatory molecules CD80 and CD86. CTLA-4 acts in an opposite manner to CD28 and downregulates the immune responses induced by T-cells [17,49,50,51]. CTLA-4 is mainly expressed on T-regulation cells and functions as an immune checkpoint, and genomic mutations on CTLA-4 have been associated with immune deficiency, leading to human autoimmune diseases [51,52]. In early previous studies, the knockout of CTLA-4 in mouse models led to fulminant lymphocytic infiltration in almost all organs, which caused the death of the animals [53]. Animal models with the knockout of CTLA-4 are important for the study of autoimmune diseases. Based on the findings of the previous studies, Allison et al. hypothesized that the transient blockade of CTLA-4 might increase the proliferation and activation of T-cells to a higher level than that which may be allowed and tolerated by the normal physiology, such that the transient blockade of CTLA-4 using an antibody could provide a new strategy for anti-cancer therapy [53,54]. In a previous preclinical study conducted by Allison et al., the inhibition of CTLA-4 using antibody blockade enhanced the anti-tumor immunity in a mouse model [55]. The results of these previous pre-clinical studies have encouraged the development of anti-CTLA-4 antibodies for use in anti-cancer therapy. Two humanized CTLA-4 antibodies, ipilimumab, and tremelimumab have been developed and tested in clinical use since 2000 [56,57,58]. 

Two large phase III clinical trials have demonstrated that ipilimumab significantly prolonged the survival of advanced melanoma patients when compared with the peptide vaccine glycoprotein 100 and standard chemotherapy with dacarbazine regimen [59,60]. In 2011, ipilimumab was approved by the U.S. FDA for clinical use in the treatment of advanced melanoma. Ipilimumab in combination with nivolumab was later approved by the U.S. FDA for the treatment of advanced renal cell carcinoma, mismatch repair deficient metastatic colorectal cancer, and unresected malignant pleural mesothelioma, based on the results of three pivotal clinical trials (CheckMate-214, CheckMate-142, and CheckMate-743) [61,62,63,64]. The efficacy of ipilimumab has also been explored in lung cancer. Small cell lung cancer (SCLC) is a subtype of lung cancer with an extremely poor prognosis, and the advances of therapy for SCLC are still limited, despite the current state of chemotherapy [64]. Ipilimumab in combination with chemotherapy or nivolumab has been investigated in several previous clinical trials (CA184-041, CA184-156, CheckMate 331, and CheckMate 451), but all these clinical trials have shown that the addition of ipilimumab failed to benefit the objective response rate, PFS, and overall survival, when compared with the results in control patients [65,66,67,68].

For advanced NSCLC, it has been theorized that combination immunotherapy may have increased antitumor activity, compared with single-agent therapy. A dual immune checkpoint inhibitor combination therapy (anti-PD-1 plus anti-CTLA-4) has been recently tested for the treatment of advanced NSCLC in clinical trials. In a previous phase 1 clinical trial (CheckMate-012), the combination of nivolumab (anti-PD-1) and ipilimumab (anti-CTLA-4) is an effective and safe therapy for advanced NSCLC patients [69]. Based on the results of the CheckMate-012 trial, an open-label phase 3 clinical trial, CheckMate-227, has been conducted to investigate the efficacy of nivolumab plus ipilimumab, compared with nivolumab alone and chemotherapy, in PD-L1 positive advanced NSCLC. The combination of nivolumab and ipilimumab led to a significantly longer duration of overall survival than chemotherapy [26,70]. Another recent phase 3 clinical trial, CheckMate 9LA, demonstrated that the combination of nivolumab and ipilimumab with two cycles of chemotherapy also led to a significant improvement in overall survival, compared to conventional chemotherapy alone [27]. In a phase 1&2 clinical trial, KEYNOTE-021, the analysis of cohorts D and H showed that pembrolizumab (anti-PD1) combined with ipilimumab therapy increased the anti-tumor activity, compared with a single agent, in pretreated advanced NSCLC patients [71]. Further study may be needed to explore the efficacy of pembrolizumab in combination with ipilimumab as a front-line therapy for NSCLC. 

Tremelimumab is another anti-CTLA-4 inhibitor that is currently under investigation in clinical trials for various cancers, e.g., melanoma, malignant pleural mesothelioma, SCLC, and NSCLC [17,53]. Durvalumab in combination with tremelimumab therapy had been explored in early-phase clinical trials, and these trials showed that this combination therapy had durable clinical activity and an acceptable safety profile in patients with pretreated and relapsed extensive-stage (ES)-SCLC patients [57,72]. Therefore, tremelimumab plus durvalumab plus chemotherapy had been tested in a pivotal phase 3 clinical trial CASPIAN. In the CASPIAN trial, the addition of tremelimumab to durvalumab plus chemotherapy did not lead to a significant improvement in overall survival [72]. The combination of tremelimumab and durvalumab for NSCLC has been tested in clinical trials; however, this combination only benefitted the survival in limited groups of patients. The ARCTIC trial showed that durvalumab, in combination with tremelimumab, leads to an improvement of overall survival in heavily pre-treated metastatic NSCLC patients, when compared with standard care [73]. The MYSTIC Phase 3 trial demonstrated that the combination of tremelimumab and durvalumab only led to a survival benefit for metastatic NSCLC patients with high blood tumor mutational burden (bTMB, ≥20 mutations per megabase) [74]. More clinical studies are still needed to explore the use of tremelimumab in combination with other treatment modalities or drugs in advanced NSCLC. 

The mechanism of anti-PD-1/PD-L1 and anti-CTLA-4 ICIs is summarized in Figure 1. The main clinical trials which have approved anti-PD-1/PD-L1 and anti-CTLA-4 ICIs for front-line NSCLC therapy are listed in Table 1.

## 3. Neoadjuvant and Adjuvant Immunotherapy in Surgically Resectable NSCLC

### 3.1. Immune Checkpoints Inhibitors (ICIs) in Neoadjuvant Therapy

Uncompleted resection by surgery is always considered in NSCLC with locally advanced disease or mediastinal lymph node metastasis (stage II and III disease), where neoadjuvant therapies, e.g., chemotherapy, radiation therapy, or concurrent chemoradiotherapy, are suggested before surgery [75,76]. Recently, ICIs have been applied and investigated for neoadjuvant therapy in NSCLC. In a previous preclinical study, Cascone et al. established a mouse model by inoculating NSCLC 344SQ-OVA+ cells into the flank of syngeneic mice, where the mice were divided into four groups to compare the efficacy of different neoadjuvant immunotherapies. The mice were treated with 3 doses of the neoadjuvant anti-PD-1 antibody, anti-CTLA-4 antibody, or anti-PD-1 plus anti-CTLA-4 antibodies, or observation, followed by surgical resection of primary tumors in all mice. The observational mice received post-surgery adjuvant therapies with anti-PD-1 antibody, anti-CTLA-4 antibody, or anti-PD-1 plus anti-CTLA-4 antibodies. The results of this study showed that either single-agent or combination neoadjuvant therapies contributed to significantly longer survival than all adjuvant therapies in the mouse model. In a subgroup analysis of mice receiving neoadjuvant therapies, the combination was significantly superior to a single agent in prolonging survival. In addition, the neoadjuvant combination therapy significantly reduced lung metastasis, when compared with a single agent and all treatment modalities, in the adjuvant setting (single and combination) [77]. Based on the promising results of this pre-clinical study, several clinical trials investigating neoadjuvant immunotherapy have been initiated [77,78]. A previous study has shown that neoadjuvant therapy with single nivolumab before surgery had a 45% major pathological response (MPR), acceptable toxicity, and no delay of surgery [79]. A previous report found that nivolumab plus ipilimumab therapy had the trend of more effective in current or former smokers than never smokers based on the results of the CheckMate 227 trial [80]. Another clinical study showed that neoadjuvant nivolumab plus ipilimumab in resectable NSCLC is feasible, and all the patients enrolled in the study were active and former smokers [81]. A previous meta-analysis review showed that neoadjuvant immunotherapy was more effective than neoadjuvant chemotherapy regarding the MPR and pathological complete response (PCR) in resectable NSCLC. In the same analysis, the surgical resection rate was also similar between neoadjuvant immunotherapy and neoadjuvant chemotherapy (88.7% vs. 70–90%) [82]. 

In a recent phase 2 clinical trial (NEOSTAR), stages I to IIIA NSCLC patients were randomized to receive neoadjuvant therapies with nivolumab alone or nivolumab plus ipilimumab, followed by surgery. In the analysis of 37 patients with surgical resection, the MPR was 24% for nivolumab alone, and 50% for nivolumab combined with ipilimumab. The NEOSTAR trial indicated that neoadjuvant therapy, with either nivolumab alone or the combination of nivolumab and ipilimumab, achieved pathological response in surgery. The results of the same trial showed that the neoadjuvant combination of nivolumab and ipilimumab produced significantly higher pathologic responses, immune infiltrations, and immunologic memory in the resected tumor than nivolumab alone [83]. Cytotoxic chemotherapy augments the immunogenicity of cancer cells by inducing antigenicity and adjuvanticity [84]. Immunogenic cell death (ICD) is associated with adaptive stress response which promotes the maturation of dendritic cells (DCs). In a lung cancer mouse model, chemotherapy promotes the ICD pathway to enhance the anti-tumor ability of anti-PD-1 and anti-CTLA4 antibodies [84,85]. In addition, chemotherapy might have off-target effects on suppressing myeloid-derived suppressor cells (MDSCs) or regulatory T (Treg) cells to stimulate anti-tumor immunity [86]. Together, these indicated that chemotherapy in combination with ICIs successfully improved the survival of metastatic NSCLC patients [24,25,27,29,30,31]. The addition of ICIs to conventional chemotherapy in neoadjuvant therapy for resectable NSCLC has been tested in two previous clinical trials. Nivolumab in combination with conventional chemotherapy as neoadjuvant therapy for resectable stage IIIA NSCLC was explored in phase 2 clinical study (NADIM), where the results of this trial showed 77.1% 24-month PFS in patients receiving tumor resection after neoadjuvant therapy [87]. Another phase 2 clinical trial investigated the efficacy of neoadjuvant atezolizumab plus chemotherapy in stage II-IIIA NSCLC. A total of thirty patients were enrolled in this phase 2 clinical trial, of which 29 finally received surgery and 17 (57%) had MPR, which was achieved with the neoadjuvant atezolizumab in combination with chemotherapy [88]. Single atezolizumab and pembrolizumab monotherapy as neoadjuvant therapy has been also tested in two previous clinical studies. Both clinical trials recruited potentially resectable stage I to III NSCLC [78,89]. Neoadjuvant single atezolizumab achieved 18% MPR in the LCMC3 clinical trial [78,89]. Ready et al. showed that neoadjuvant single pembrolizumab had 28% MPR in the other phase 2 clinical trial [83]. 

There are remaining some early-stage NSCLC patients who do not receive surgery because of reasons including poor cardiopulmonary reserve, extremely old age, poor performance status, and personal refusal. Therefore, radiotherapy such as stereotactic ablative radiotherapy (SABR) can be an alternative treatment for early-stage NSCLC patients who are unable to receive surgery [90,91]. Previous studies had shown that local radiation therapy can stimulate the release of tumor-associated antigens (TAAs) and damage-associated molecular patterns (DAMPs). The TAAs and DAMPs promote immune cell priming and destruct immunosuppressive tumor-supporting stroma and these result in the enhancement of the anti-cancer effect of ICIs in NSCLC [85,92]. The efficacy of ICIs enhanced by radiotherapy is also called the abscopal effect [85,92], and compatible with the promising results shown in the PACIFIC trial. Using the combination of local radiation therapy and ICIs to improve local control and survival in early-stage NSCLC is warranted in future clinical trials. The results of trials using immunotherapy, with or without chemotherapy, as neoadjuvant therapy in surgically resectable NSCLC patients are summarized in Table 2. 

At present, several ongoing clinical trials are investigating the use of ICIs with or without chemotherapy as neoadjuvant therapy in resectable NSCLC (Table 3.). Several previous early-phase (phases I & II) had shown that ICIs with or without chemotherapy were feasible and effective as a neoadjuvant therapy before surgery [78,79,80,81,82,83,84,85,86,87,88,89,90,91,92,93,94,95,96,97]. Therefore, four main phase III clinical trials (KEYNOTE 617, CheckMate 816, IMpower 030, AEGEAN) are conducted and ongoing now. All four trials enrolled control groups, and explore the consolidation ICIs therapy after surgery. These four clinical trials are expected to be completed in 2024 [78,98,99,100]. 

### 3.2. Immune Checkpoint Inhibitors (ICIs) in Post-Operation Adjuvant Therapy

Some early-stage NSCLC patients receive surgical resection without neoadjuvant therapy, and post-operation adjuvant chemotherapy is generally recommended for those with high risks of recurrence [84]. The risks of post-operation recurrence in NSCLC include lymph node metastases, the main tumor size being larger than 4 cm, and extensive local invasion [101]. The use of anti-PD-1/PD-L1 ICIs with or without chemotherapy as post-surgery adjuvant therapy in NSCLC is under investigation, and no mature study result is available to date [78,102]. There are four ongoing phase 3 clinical trials considering anti-PD-1/PD-L1 ICIs for early-stage NSCLC patients after receiving complete tumor resection (ANVIL, PEARLS, IMpower010, and BR31) [78,102,103,104]. The details of these four clinical trials are summarized in Table 3. The four phase 3 clinical trials are planning to recruit about 4600 NSCLC patients receiving surgery, and are expected to be completed between 2024 and 2027. Disease-free survival (DFS) is the main primary endpoint of all four trials [78,102,103,104]. The results of these phase 3 clinical trials may bring a substantial impact on the clinical practice of NSCLC patients receiving complete resection in the future. Though the design of the four ongoing trials is similar, there is little difference among the 4 ongoing trials. First, post-operative platinum-based chemotherapy before randomized to atezolizumab or best supportive care group is a required treatment for participants of the IMpower010 trial whether post-operative chemotherapy is optional for the participants of the other 3 ongoing trials. Second, the patients in the control group of IMpower010 and ANVIL trials receive the best supportive care or observation, and the patients in the control group of the other 2 ongoing trials (PEARLS and BR31) receive placebo [78,102,103,104]. Patients in the BR31 trial would have the tests EGFR mutation and ALK rearrangement for further subgroup analysis. Patients with EGFR mutation or ALK rearrangement would be excluded from the ANVIL trial. The tests of EGFR mutation and ALK rearrangement are not mandatory in PEARLS and IMpower010 trials. All the trials have the test of tumor tissue PD-L1 expressions for further subgroup analysis in the future [78,102,103,104]. The results of the four ongoing trials will provide information on ICIs with or without chemotherapy as post-operative adjuvant therapy for clinical practice. 

### 3.3. Immune Checkpoint Inhibitors (ICIs) in Neoadjuvant Therapy or Adjuvant Therapy

According to the results of a pre-clinical study by Cascone et al., the neoadjuvant ICIs seem to contribute better survival benefits than the adjuvant setting has in the mouse model [77]. The complete clinical trials showed that ICIs with or without chemotherapy as neoadjuvant therapy achieved pathological response and contribute to complete surgical resection. However, there were remaining some NSCLC patients receiving neoadjuvant therapy who did not receive surgical resection finally because of complication or disease progression in neoadjuvant therapy [78,79,83,87,88,89,93,94]. In the main four ongoing trials investigating ICIs as adjuvant therapy for post-surgery NSCLC patients, the enrolled patients were required to have complete surgical resection (R0) [78,102,103,104]. However, some NSCLC patients have incomplete surgical resection in real-world clinical practice [105]. Post-operative adjuvant therapy such as chemotherapy and radiotherapy are suggested for incomplete resection NSCLC patients. However, the survival benefit of post-operative conventional chemotherapy and radiotherapy is limited for incomplete resection NSCLC patients, and the prognosis of these patients are not well [106]. ICIs in addition to chemotherapy or radiotherapy may provide survival benefits for incomplete resection NSCLC patients, but the 4 ongoing trials of adjuvant ICIs cannot answer this clinical query [78,102,103,104]. 

In currently ongoing four main phase III clinical trials (KEYNOTE 617, CheckMate 816, IMpower 030, AE-GEAN) with neoadjuvant chemotherapy plus ICIs or placebo, post-operative consolidation ICIs therapy is administrated in the treatment group patients [78,98,99,100]. These four clinical trials will provide clear evidence on the efficacy of ICIs administrated before and after surgery in early-stage and resectable NSCLC. 

## 4. Therapeutic Vaccination in NSCLC

### 4.1. Cancer Vaccines and Tumor Antigens

Unlike normal vaccination, cancer vaccines are introduced either for early cancer prevention or during cancer treatment (i.e., preventive or therapeutic cancer vaccines, respectively). Cancer vaccines are designed based on tumor antigens, to trigger cytotoxic T-cell responses and improve immunosurveillance for tumor cells. Tumor antigens can be classified as tumor-associated antigens and tumor-specific antigens. Tumor-associated antigens are self-antigens and can be expressed in a subset of normal host cells. They are generally characterized by low immunogenicity, and T-cells have low-affinity receptors, that are unable to mediate effective anti-tumor immune responses. Additionally, the T-cells that recognize these antigens may be removed from the immune repertoire through central and peripheral tolerance mechanisms. Tumor-specific antigens (or tumor neoantigens) are strictly specific to cancer cells and are not expressed on the surface of normal cells. Tumor-specific antigens elicit high-affinity T-cells and are less likely to be deleted by central and peripheral tolerance [107,108]. Emerging evidence has suggested that neoantigens play a pivotal role in tumor-specific T-cell-mediated antitumor immunity and immunotherapy or vaccines targeting tumor-specific antigens should theoretically be less likely to induce autoimmunity. Many tumor-associated antigens and tumor-specific antigens have been identified. Targeting these antigens is important for cancer immunotherapy, and several clinical trials investigating different tumor types are now underway [107].

Platforms for cancer vaccines are categorized as a cellular, viral vector, or molecular (peptide, DNA, or RNA), and recent therapeutic cancer vaccines have shown advances in novel platforms and tumor-specific antigens. Cancer vaccines enhance the tumor-specific T-cell response, with related research focusing on vaccine technologies, delivery vaccine platforms, and more immunogenic antigen selection (e.g., predicated on neoantigens), which could amplify and broaden the endogenous repertoire of tumor-specific T-cells, to enhance the anti-tumor activity [107]. Sipuleucel-T (Provenge^®^; Dendreon) was the first therapeutic autologous dendritic cell-based cancer vaccine approved by the U.S. FDA (in 2010) for the treatment of asymptomatic or minimally symptomatic metastatic castration-resistant prostate cancer with tolerable safety, which prolonged median survival by 4.1 months, compared with the results in those treated with placebo [109]. The second U.S. FDA-approved and first oncolytic virus vaccine was talimogene laherparepvec (T-VEC) (Imlygic^®^), which has been indicated for the local treatment of unresectable cutaneous, subcutaneous, and nodal lesions in patients with melanoma recurrent after initial surgery, and which has demonstrated a significantly durable response rate [110,111,112].

### 4.2. Interaction between Immune Cells and Tumor Microenvironment

The tumor microenvironment is characterized by the predominance of immunosuppression [113]. The complex and dynamic nature of the interactions between immune cells and the tumor microenvironment could influence tumor growth, invasion, and metastasis [113]. The interactions consisting of cellular components including various myeloid and lymphoid cells, fibroblasts, and endothelial cells that via direct interactions or biochemical cues (auto-, para-, and endocrine signaling) to communicate with tumor cells. Non-cellular components consisting of extracellular matrix, mechanical pressure, and tumorigenic conditions like acidity, hypoglycemia, and hypoxia that impact tumor behavior. These components are essential to stimulate the heterogeneity of tumor cells, clonal evolution and increase the resistance leading to tumor progression and metastasis [113,114]

The fate of a tumor is dependent on the dynamic properties of the anti- to protumorigenic tumor microenvironment. The antitumorigenic tumor microenvironment contains normal fibroblasts, dendritic cells, natural killer (NK) cells, cytotoxic T cells, and M1-activated tumor-associated macrophages involving the activity of pro-inflammatory cytokines. The protumorigenic tumor microenvironment contains immunosuppressive effects of M2-activated tumor-associated macrophages involving the production of anti-inflammatory cytokines, myeloid-derived suppressor cells, regulatory T cells and B cells, cancer-associated fibroblasts producing aberrant extracellular matrix, TIE2-expressing monocytes, and mast cells with angiogenesis stimulatory activity. Similar to tumor-associated macrophages, neutrophils and T helper cells can have both pro- and antitumorigenic activity depending on tumor and immune context [114].

Activation of the immune system to combat cancer was an appealing method for decades. However, the tumor microenvironment including immunosuppressive immune cells certainly contributes to hamper immunotherapy. Any therapy aiming to reduce immunosuppression must be given simultaneously to other agents that activate immune reactions or counteract other suppressive arms such as the myeloid cell compartment.

Immune activating therapies included different treatment modalities such as tumor-peptide-based vaccines, oncolytic viruses, agonistic antibodies, and immune cell therapies based on T cells, NK cells, and dendritic cells. Options to reduce the action of immunosuppressive cells or molecules were evaluated. For example, antibodies that block chemokine receptors may prevent the accumulation of suppressive myeloid cells such as M2 macrophages in the tumor microenvironment, the small-molecule mebendazole polarizes macrophages toward the anti-tumoral M1 phenotype and common chemotherapeutics such as gemcitabine, or tyrosine kinase inhibitors, reduces both MDSCs and/or regulatory T cells in patients. Such agents may be of high interest to combine with ICIs and activating immunotherapeutics. Agents affecting angiogenesis may be of interest since they limit blood supply and normalize the dysregulated blood vessels in the tumor tissues. Such normalization may allow for better lymphocyte attachment, rolling, and transmigration into the tumor sites [115].

Preventive cancer vaccines administrated at pre-malignant stages of disease (i.e., before tumor-associated immunosuppression is established) may be feasible. Hepatitis B virus (HBV) and human papillomavirus (HPV) vaccines are well-known and very effective in preventing the initial infection and, subsequently, decreasing the risk of cancer formation caused by these viruses (hepatocellular carcinoma by HBV; cervical cancer and HPV-positive oral cancers by HPV). The local tumor microenvironment at these stages exerts less immunosuppression, and preventive cancer vaccines targeting candidate tumor antigens identified from biopsies with sequencing, administered at earlier stages, could strengthen immunosurveillance, preventing the recurrence or slowing the progression of cancer, thus achieving immunoprevention [111]. Future research may focus on preventive cancer vaccines for healthy persons, with or without a known genetic risk of cancer, for cancer prevention.

Although many therapeutic cancer vaccines have been extensively designed and investigated to treat advanced cancer, their clinical effectiveness has been profoundly unsatisfactory to date. The reasons for such limited efficacy are that the advanced lesion may already induce multiple types of the immunosuppressive tumor microenvironment, the complexity of identifying tumor-specific antigens, and as most cancer vaccines targeting tumor antigens are non-mutated overexpressed self-antigens, eliciting mainly T-cells with low-affinity T-cell receptors, which are unable to mediate an effective anti-tumor response [108,111]. Combination therapy with therapeutic cancer vaccines, anti-angiogenic therapies, and/or *immune* checkpoint inhibitors can provide a potentially synergistic effect to enhance the immune response against tumors, overcome the immunosuppressive tumor microenvironment, prevent disease recurrence, and achieve clinical efficacy in long-term cancer treatment [107,112]. With the advent of next-generation sequencing, personalized therapeutic cancer vaccines based on tumor-specific neoantigens from tumor biopsies have been emerging. Most clinical trials concerning such technology are ongoing; however, considerable uncertainty remains as to which platform will perform best. 

### 4.3. Clinical Trials of Cancer Vaccines in NSCLC

Cancer vaccines may have an adjuvant role in surgically resectable and unresectable NSCLC by consolidating responses to definitive medical, surgical, or multimodality therapy. For NSCLC, CIMAvax-EGF is a therapeutic cancer vaccine developed entirely in Cuba. The first clinical studies started in 1995, and it was licensed by the Cuban Regulatory Agency in 2008, such as a switch maintenance treatment for patients with stage IIIB/IV NSCLC after first-line chemotherapy, which was the first registration of a lung cancer vaccine in the world. The CIMAvax-EGF vaccine consists of a human recombinant EGF chemically conjugated to P64K, a recombinant carrier protein derived from Neisseria Meningitidis B, with Montanide ISA51 as an adjuvant. CIMAvax-EGF administration induces antibodies against EGF, prevents EGF from binding to its receptor, and, therefore, inhibits tumor growth. CIMAvax-EGF has been proved to be safe and immunogenic in the treatment of patients with advanced NSCLC in several clinical trials; however, the responses to vaccines were heterogeneous [116]. A phase III clinical trial of CIMAvax-EGF as switch maintenance was well-tolerated and significantly increased the median survival time of advanced NSCLC patients that completed induction vaccination, where the baseline EGF concentration predicted the survival benefit [117]. The combination of CIMAvax-EGF with ICIs may provide a potential therapeutic option for advanced lung cancer in the future, to ensure better tumor control, in terms of the good safety and immunogenic profile of CIMAvax-EGF. An ongoing clinical trial is being conducted to investigate the efficacy of this combination (Clinical Trial number: NCT02955290).

Cancer vaccination can serve as a potential therapeutic modality for cancer prevention in surgically resectable early-stage NSCLC, where more future clinical studies are warranted to determine the role of vaccination in surgically resectable early-stage NSCLC. The clinical trials (ongoing and completed) investigating cancer vaccination in surgically resectable early-stage NSCLC are summarized in Table 4. Clinical trials of preventive vaccines for lung cancer had been conducted. Early phase I trial was conducted to evaluate the effect of CIMAvax-EGF on the prevention of lung cancer development in high-risk patients, like family history of lung cancer or chronic pulmonary obstructive disease, or recurrence in patients with stage IB to IIIA NSCLC (Clinical Trial number: NCT04298606). Another phase I trial studies the side effects and how well peptide vaccine works in preventing lung cancer in high-risk groups of current and former smokers (Clinical Trial number: NCT03300817). For clinical trials of vaccines in patients with surgically resectable NSCLC, a nonrandomized pilot study reported early clinical experience with vaccine 1650-G, an allogeneic cellular vaccine using granulocyte-macrophage colony-stimulating factor as an adjuvant. 1650-G is safe, biologically reliable, and comparatively inexpensive, and could generate a robust and unequivocal immunological response. The relative frequency and kinetics of the response appear similar to that achieved with dendritic cell vaccines [118]. Although therapeutic efficacy is unknown, immature dendritic cell vaccine preparation, pulsed with apoptotic tumor cells, has similar biologic efficacy to autologous dendritic cell vaccines matured with dendritic cell/T cell-derived maturation factor-matured preparation in NSCLC patients [119]. The MAGE-A3 protein is expressed in approximately 35% of patients with resectable NSCLC patients. A double-blind, randomized, placebo-controlled phase II study investigated recombinant MAGE-A3 protein combined with an immunostimulant in completely resected MAGE-A3–positive stage IB to II NSCLC patients. Although there was no statistically significant improvement in the disease-free interval and overall survival, postoperative MAGE-A3 immunization proved to be feasible with minimal toxicity [120]. Human telomerase reverse transcriptase (hTERT) is an antigen that may represent a target for a novel anti-cancer strategy. A pilot, phase I study evaluated a prime-boost immunization regimen based on V935 (an adenoviral type 6 vector vaccine expressing a modified version of hTERT), administered alone or in combination with V934 (a DNA plasmid that expresses the same antigen), and the results revealed that the safety and feasibility of V934/V935 hTERT vaccination in cancer patients with solid tumors, including NSCLC [121]. The clinical trials (ongoing and completed) investigating cancer vaccination in surgically resectable early-stage NSCLC are summarized in Table 5.

### 4.4. Cancer Vaccination in Combination with Immunotherapies

Some cancer patients do not benefit from immunotherapy, and this low response rate may be related to a limited specific T cell response developed against tumor cells, especially for tumors with a low mutational burden. The efficacy of immunotherapy appears to depend on preexisting intratumoral CD8+ T cells underlining the necessity to induce these cytotoxic T cells with vaccination. Cancer vaccines can generate tumor-specific T cells in the periphery or in situ tumors and can drive activated peripheral T cells into the tumor microenvironment leading to increased tumor-infiltrating lymphocytes. Moreover, vaccine-mediated tumor cell death leads to the release of more cascade antigens and induces stronger immune responses specific to antigens (i.e., antigen cascade or epitope spreading) [122]. Whereas ICIs boost inactivated responses of effector T cells, vaccination can potentially activate naive T cells with tumor specificity and in this way broaden the tumor-specific immune responses. Therefore, combining cancer vaccines, which can elicit specific T cell responses, with ICIs represent attractive therapeutic options [123,124].

## 5. Future Perspectives: The Challenges of Immunotherapy in Surgically Resectable NSCLC

### 5.1. Imaging Evaluation of Response before Surgery and Post-Operative Follow-Up

The ICIs treatment has distinctive response patterns and adverse events related to the biological mechanism of anticancer activity, and the imaging evaluation of ICIs treatment is a challenge. Pseudoprogression is a phenomenon described that radiological image showed tumor progression with enlargement and/or appearance of new lesions initially, and followed by stabilization or regression of tumor without additional treatment [125,126,127]. The causes of pseudoprogression may result from the ongoing tumor growth before the achievement of the immune response, the cytotoxic T lymphocytes recruitments, and the inflammatory process surrounding the tumor micro-environment. The incidence of pseudoprogression in NSCLC with ICIs treatment is about 5% [125,126,127]. If pseudoprogression occurs in neoadjuvant therapy, it may affect the judgment and decision of surgery. Careful evaluation by using multiple images including radiological plain film, sonography, computed tomography (CT) scan, magnetic resonance imaging (MRI), and fluorodeoxyglucose (FDG)-positron emission tomography (PET) before surgery should be considered. In patients receiving post-operative adjuvant therapy with ICIs, pseudoprogression should be alert because it may mislead some patients in response to the therapy. Procedures including needle aspiration, core tissue biopsy, and/or surgery on tumors or new lesions may be needed to determined pseudoprogression or true progression in these patients. 

### 5.2. Predictive Biomarkers in Therapy

The issue of looking for biomarkers to predict the efficacy of immunotherapy in NSCLC emerges, but there is no idea predictive biomarker of immunotherapy for the complexity and diversity in tumor immune environment [9,128]. PD-L1 expression and bTMB levels were reported to correlate with response rate and PFS of anti-PD-L1 ICIs in the treatment of advanced NSCLC [128,129]. Pretreatment PD-L1 expression and TMB levels were also tested in neoadjuvant ICIs trials of early-stage NSCLC including NEOSTAR, LCMC3, NADIM and Forde et al. trials [78,79,83,87,89,130]. The MPR was positively and significantly associated with the PD-L1 expression level in the NEOSTAR trial but was not significantly associated with the PD-L1 expression level in LCMC3, NADIM, and Forde et al. studies [78,79,83,87,89,130]. TMB was associated with MPR in the study of Forde et al. but was not observed in the LCMC3 trial [78,79,89]. Heterogenous associations between PD-L1 expression, TMB levels and MPR were observed in these clinical trials [78,79,83,87,89,130]. The PD-L1 expression and TMB levels should be carefully and interpreted before and during neoadjuvant therapy. 

In adjuvant ICIs setting, the association between PD-L1 expression level and the outcome will be analyzed and shown after the four ongoing trials completed in the future [78,102,103,104]. 

Previous studies reported hyperprogressive disease (HPD) in NSCLC treated with ICIs [Choi, Kim, Chen]. HPD was characterized by rapid tumor progression after the initiation of ICIs therapy and was associated with a poor prognosis [131,132,133]. In several previous clinical analyses, the risk factors associated with HPD in NSCLC with ICIs treatment are serum lactate dehydrogenase (LDH) > upper limit of normal, neutrophil-to-lymphocyte ratio (NLR) of ≥3, liver metastasis, and > 2 metastatic sites [132,133,134]. In early-stage resectable NSCLC receiving neoadjuvant and/or adjuvant ICIs, the pretreatment LDH and NLR should be tested and imaging evaluation should be closely followed in a patient with increased risk of HPD. 

Previous clinical studies have shown that early-stage NSCLC harboring EGFR mutations had an increased risk of recurrence and metastasis [135,136]. A recent study has demonstrated that consolidation with durvalumab (anti-PD-L1 inhibitor) did not benefit the survival in stage III NSCLC patients with EGFR mutation receiving induction-concurrent chemoradiation therapy. Consolidation therapy with durvalumab also increased immune-related adverse events in post-chemoradiotherapy EGFR-mutated stage III NSCLC [137]. To date, osimertinib is the recommended post-operative adjuvant therapy for stages IB–IIIA NSCLC with EGFR mutation, based on the promising results shown in the ADAURA trial [8]. Alectinib is an effective and safe targeted therapy for advanced NSCLC patients with ALK rearrangement [138,139]. The use of alectinib as a post-surgery adjuvant therapy in stage IB–IIIA ALK-mutated NSCLC patients is currently being explored in a phase III clinical trial (ALINA trial, NCT03456076) [140]. For, EGFR- and ALK- mutated early-stage NSCLC, surgery followed by adjuvant TKIs may be considered priorly to ICIs. The other oncogene-addicted NSCLC (ex. BRAF, MET, HER2, RET, K-RAS, and NTRK mutations), neoadjuvant and adjuvant ICIs can be considered for resectable NSCLC unless the appearance of more effective and safer target therapy than ICIs. 

### 5.3. The Primary Endpoints: Pathological Response and Survival; Timing and Type of Surgery

MPR was the endpoint employed most frequently in early-phase trials with neoadjuvant ICIs [78,79,80,81,82,83,84,85,86,87,88,89,90,91,92,93,94,95,96,97]. For early-stage NSCLC, the treatment goal should focus on disease cure and prolonging survival. Therefore, the event-free survival (EFS) and OS are employed as a primary endpoint by the recent ongoing trials (KEYNOTE 617, CheckMate 816, IMpower 030, AEGEAN) [78,98,99,100]. Timing of surgery is also a concerning issue in using ICIs as neoadjuvant therapy. In the design of clinical trials with neoadjuvant ICIs, the neoadjuvant therapies were administrated for 2–4 cycles [78,79,80,81,82,83,84,85,86,87,88,89,90,91,92,93,94,95,96,97,98,99,100,101,102]. These studies reported that the radiological response assessed by Response Evaluation Criteria in Solid Tumors (RECIST) did not correlate with pathological response [78,79,80,81,82,83,84,85,86,87,88,89,90,91,92,93,94,95,96,97,98,99,100,101,102]. For example, Forde et al. showed that 90% of patients with pathological response in the study had the radiological stable disease before surgery [79]. In NEOSTAR, 11% of study patients were found to had the phenomenon of pseudoprogression in mediastinal lymph nodes radiologically, and no cancer cells were found in these lymph nodes pathologically in surgical resection [83]. Progressive disease in neoadjuvant therapy which lead to the abandonment of surgery was also reported in these trials [78,98,99,100]. In the LCMC3 trial, 10 study patients were found inoperable in exploration or experienced disease progression in neoadjuvant atezolizumab [78,89]. Regarding the HPD of ICIs treatment, NSCLC patients with risk factors may be considered surgery directly followed by adjuvant therapy. Fortunately, no unexpected immune-related adverse event (irAE) that lead to the delay of surgery was reported in these trials with neoadjuvant ICIs in early-stage NSCLC [78,79,80,81,82,83,84,85,86,87,88,89,90,91,92,93,94,95,96,97,98,99,100,101,102]. In an early trial of neoadjuvant therapy with the combination of nivolumab and ipilimumab in melanoma, 18 of 20 study patients (90%) experienced grade 3 and 4 irAE, and cessation of therapy was required in most patients (17 of 18) [141]. 

Regarding the surgery type, extensive resections such as pneumonectomy in early-stage NSCLC contributed unfavorable outcomes and severer morbidity compared with lobectomy [142,143]. The pathological responses are achieved in part NSCLC patients receiving neoadjuvant ICIs, and surgery with more pulmonary volume may be considered and explored in future studies. 

Cancer vaccination is mainly explored in post-resected NSCLC for the prevention of recurrence. The comparison of efficacy between post-operative ICIs and vaccination is not clear. Vaccination may be an alternative choice for post-operative NSCLC patients who ever experience irAE in neoadjuvant therapy or at high risk of irAE. 

The optimal strategy of pretreatment evaluation, neoadjuvant therapy, surgery, and adjuvant therapy is summarized in Figure 2. 

## 6. Conclusions

Immunotherapies, including ICIs and vaccination, have potential effects in reducing recurrence and prolonging survival for surgically resectable NSCLC patients. The exploration of optimal patient selection and treatment modality combinations for immunotherapy in early-stage resectable NSCLC provides further direction for future studies.

## Figures and Tables

**Figure 1 vaccines-09-00689-f001:**
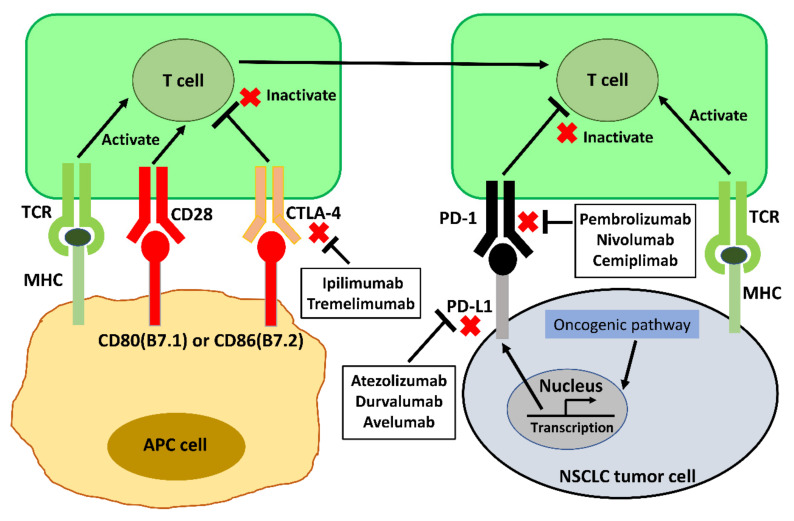
Mechanism of anti-PD-1/PD-L1 and anti-CTLA-4 ICIs in anti-cancer therapy for NSCLC. Abbreviations: APC, antigen-presenting cell; TCR, T-cell receptor; MHC, major histocompatibility complex; PD-L1, programmed Death-Ligand 1; PD-1, programmed Cell Death Protein-1 (PD-1); CTLA-4, cytotoxic T-lymphocyte-associated protein 4.

**Figure 2 vaccines-09-00689-f002:**
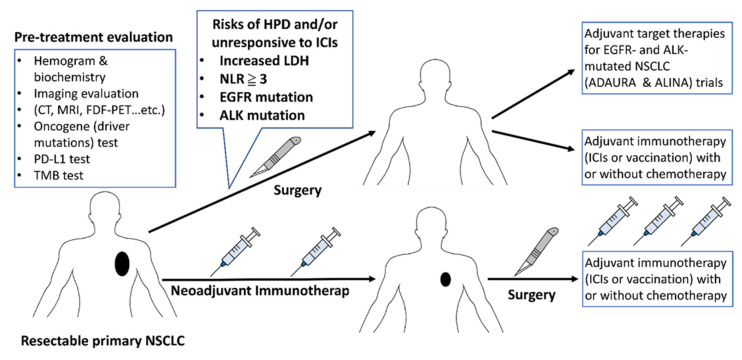
To optimize the strategy of pretreatment evaluation, neoadjuvant therapy, surgery, and adjuvant therapy are for early-stage NSCLC. Abbreviations: CT, Computed Tomography; MRI, Magnetic Resonance Imaging; FDG-PET, fluorodeoxyglucose-positron emission tomography; PD-L1, programmed Death-Ligand 1; TMB, tumor mutation burden; HPD, hyperprogressive disease; LDH, lactate dehydrogenase; NLR, neutrophil-to-lymphocyte ratio; EGFR, epidermal growth factor receptor; ALK, anaplastic lymphoma kinase; NSCLC, non-small cell lung cancer; ICIs, immune checkpoint inhibitors.

**Table 1 vaccines-09-00689-t001:** Summary of main clinical trials approving anti-PD-1/PD-L1 and anti-CTLA-4 ICIs as front-line NSCLC therapy.

Trial Name [Reference]	Histology	PD-L1 Expression	Arm 1	Arm 2	Median OS (Months) Arm 1 vs. Arm 2 (HR)
KEYNOTE-024 [22]	NSCLC	PD-L1 positive (≥50% expression)	Pembrolizumab	Platinum-based chemotherapy	30.0 vs. 14.2 0.63
KEYNOTE-042 23]	NSCLC	PD-L1 positive (≥1% expression)	Pembrolizumab	Platinum-based chemotherapy	16.7 vs. 12.1 0.81
KEYNOTE-189 [24]	Non-squamous	Any level	Pembrolizumab + pemetrexed + cisplatin/ carboplatin	Placebo + pemetrexed + cisplatin/ carboplatin	22.0 vs. 10.7 0.56
KEYNOTE-407 [25]	Squamous	Any level	Pembrolizumab + carboplatin + paclitaxel/nab–paclitaxel	Placebo + carboplatin + paclitaxel/nab–paclitaxel	17.1 vs. 11.6 0.71
CHECKMATE-227 [26]	NSCLC	Any level	Nivolumab + ipilimumab	Platinum-based chemotherapy + pemetrexed (non-squamous)/ gemcitabine (squamous)	PD-L1 ≥ 1% 17.1 vs. 14.9 0.79 PD-L1 negative 17.2 vs. 12.2 0.62
CHECKMATE 9LA [27]	NSCLC	Any level	Nivolumab + ipilimumab + 2 cycles of platinum-based chemotherapy + pemetrexed (non-squamous)/ paclitaxel (squamous)	Platinum-based + pemetrexed (non-squamous) paclitaxel (squamous)	15.6 vs. 10.9 0.66
EMPOWER-Lung 1 [28]	NSCLC	PD-L1 positive (≥50% expression)	Cemiplimab	Platinum-based chemotherapy	NR vs. 14.2 0.57
IMpower150 [29,30]	Non-squamous	Any level	Atezolizumab + bevacizumab + carboplatin + Paclitaxel	Bevacizumab+ carboplatin + Paclitaxel	19.8 vs. 14.9 0.76
IMpower130 [31]	Non-squamous	Any level	Atezolizumab + Carboplatin + nab-paclitaxel	Carboplatin + nab-paclitaxel	18.6 vs. 13.9 0.79
IMpower110 [32]	NSCLC	PD-L1 positive (≥1% expression)	Atezolizumab	Platinum-based chemotherapy	20.2 vs. 13.1 0.59
PACIFIC [33,34]	NSCLC	Any level	Durvalumab	Placebo	47.5 vs. 29.1 0.71

Abbreviations: NSCLC, non-small cell lung cancer; ICIs, immune checkpoint inhibitors; PD-L1, programmed Death-Ligand 1; PD-1, programmed Cell Death Protein-1 (PD-1); CTLA-4, cytotoxic T-lymphocyte-associated protein 4.

**Table 2 vaccines-09-00689-t002:** Results of clinical trials using immunotherapy with or without chemotherapy as neoadjuvant therapy for resectable NSCLC patients.

Trial [Reference]	Stage	Number of Patients Recruited	Drugs Used in Neoadjuvant Therapy	Primary Endpoint	MPR (%)
Forde et al. (NCT02259621) [79]	Stages I-IIIA	21	Nivolumab (monotherapy)	Safety and feasibility	45
NEOSTAR (NCT03158129) [83]	Stages IA-IIIA	44	Nivolumab or nivolumab + ipilimumab	MPR	24 in nivolumab group 50 in nivolumab + ipilimumab group
NADIM (NCT03081689) [87]	Stages IIIA	46	Nivolumab + carboplatin + paclitaxel	24-month PFS (77.1%)	83
Shu et al. (NCT02716038) [88]	Stages II-IIIA	30	Atezolizumab + carboplatin + nab-paclitaxel	MPR	57
LCMC3 (NCT02927301) [78,89]	Stages IB-IIIB	82	Atezolizumab (monotherapy)	MPR	18
Ready et al. (NCT02818920) [89]	Stages IB-IIIB	25	Pembrolizumab (monotherapy)	MPR	28
PRINCEPS (NCT02994576) [93]	Stages IA (>2 cm)-IIIA	30	Atezolizumab (monotherapy)	Toxicity	Not available
Gao et al. (ChiCTR-OIC-17013726) [94]	Stages IA-IIIB	40	Sintilimab (monotherapy)	MPR	40.5

Abbreviations: MPR, major pathological response; PFS, progression-free survival.

**Table 3 vaccines-09-00689-t003:** Ongoing clinical trials using immunotherapy with or without chemotherapy as neoadjuvant therapy for resectable NSCLC patients.

Trial [Reference]	Phase	Stage	Number of Patients Recruited or Target Number	Drugs Used in the Trial	Primary Endpoint
NEOMUN (NCT03197467) [95]	II	Stages II-IIIA	30	Pembrolizumab (monotherapy)	Safety and feasibility
IFCT-1601 IONESCO (NCT03030131) [96]	II	Stages IB (>4 cm)-IIIA	50	Durvalumab (monotherapy)	Complete surgical resection (R0)
ACTS-30 (NCT03694236) [97]	Ib	Resectable Stage IIIA	14	Durvalumab + chemoradiotherapy	Safety and feasibility
KEYNOTE 617 (NCT03425643) [78]	III	Stages II-IIIB	786	Chemotherapy + pembrolizumab/placebo × 4 cycles → surgery → pembrolizumab/placebo × 13 cycles	Event-free survival (EFS) and OS
CheckMate 816 (NCT02998528) [98]	III	Stages IB-IIIA	350	Chemotherapy + nivolumab × 3 cycles vs. chemotherapy alone × 3 cycles → surgery	EFS and pathological complete response (pCR)
IMpower 030 (NCT03456063) [99]	III	Stages II-IIIB	374	Chemotherapy + atezolizumab/placebo × 4 cycles → surgery → pembrolizumab/placebo × 16 cycles	MPR, EFS
AEGEAN (NCT03800134) [100]	III	Stages IIA-IIIB	300	Chemotherapy + durvalumab/placebo × 3 cycles → surgery → durvalumab/placebo × 12 cycles	MPR

Abbreviations: EFS, even-free survival; pCR, pathological complete response; MPR, major pathological response.

**Table 4 vaccines-09-00689-t004:** Ongoing clinical trials using ICIs as adjuvant therapy for post-surgery NSCLC patients.

Trial	Stage	Estimated Enrollment	Treatment Procedure	Primary Endpoint
ANVIL (NCT02595944)	Stages IB-IIIA	903	Surgery +/− chemotherapy → nivolumab vs. observation	DFS, OS
PEARLS (NCT02504372)	Stages IB-IIIA	1177	Surgery +/− chemotherapy → pembrolizumab vs. placebo	DFS
IMpower010 (NCT02486718)	Stages IB-IIIA	1280	Surgery +/− chemotherapy → atezolizumab vs. best supportive care	DFS
BR31 (NCT02273375)	Stages IB-IIIA	1360	Surgery +/− chemotherapy → durvalumab vs. placebo	DFS

Abbreviations: ICIs, immune checkpoint inhibitors; NSCLC, non-small cell lung cancer; DFS, disease-free survival; OS, overall survival.

**Table 5 vaccines-09-00689-t005:** Overview of cancer vaccination in surgically resectable, non-advanced, or preventions of non-small cell lung cancer.

Vaccine	Aims and Design	Indications	Intervention or Results	Trial Phase and Status	Clinicaltrials.gov Identifier
CIMAvax-EGF	Prevention of lung cancer development or recurrence Human recombinant EGF coupled to a carrier protein, recombinant P64K	NSCLC, stage IB to IIIA	Recombinant Human EGF-rP64K/Montanide ISA 51 Vaccine I Loading phase: 0, 2, 4, and 6 weeks, and maintain phase: Q4W in the absence of disease progression or unacceptable toxicity	Early phase I, recruiting	NCT04298606
1650-G Vaccine	Allogeneic cellular vaccine	NSCLC, stages I and II after completion of initial definitive therapies	Administered intradermally in the thigh at week 0 and week 4	Phase II, completed; 12 participants	NCT00654030 [96]
Tumor vaccine	Neoantigen tumor vaccine	Lung adenocarcinoma, stage IIIA, post radical operation	5 injections for every 3 days and then 1 injection for every 3 months until recurrence or up to 2 years	Phases I and II, not yet recruiting	NCT03807102
Autologous dendritic cell cancer vaccine	Autologous dendritic cell cancer vaccine	NSCLC, stages I to III	Injection under the skin in the front, upper thigh. Two vaccine injections total, given one month a part.	Phases II, completed, 32 participants	NCT00103116 [97]
MCU1 vaccine	MUC1 peptide-Poly-ICLC vaccine	Current and former smokers at high risk for lung cancer.	Subcutaneously at weeks 0, 2, and 10.	Phase II, recruiting	NCT03300817
MUC1 (Mucin1) peptide vaccine	Vaccine + PolyICLC	NSCLC, stages IA to IIIB	Subcutaneously every 3 weeks × 3	Phases I and II, recruiting	NCT01720836
Autologous tumor cell vaccine	Autologous tumor Lysate-Pulsed Dendritic Cells vaccine	NSCLC, stages IB to IIIA	SubcutaneouslyTwice subcutaneously twice, 4 weeks apart	Phase I, completed	NCT00023985
MIDRIXNEO	Autologous neoantigen-targeted dendritic cell vaccine	NSCLC, considered functionally operable and surgically resectable	Intravenous infusions of MIDRIXNEO-LUNG DCs every 2 weeks, using an intra-patient dose escalation scheme	Phases I and II, recruiting	NCT04078269
DRibble vaccine	Cyclophosphamide with Dribble vaccine alone or with GM-CSF or Imiquimod for adjuvant treatment	NSCLC, stages IIIA or IIIB	Cyclophosphamide is administered as a single dose three days prior to vaccine therapy.	Phase II, completed	NCT01909752
Peptide-pulsed DCs	Mutant p53 peptide-pulsed dendritic cell vaccine	NSCLC, stages IIIA or IIIB with p53 mutation	Four timesIV weekly for 5 weeks.	Phase II, completed	NCT00019929
Monoclonal antibody 11D10/3H1 anti-idiotype vaccine	Combining vaccine therapy with radiation therapy	NSCLC, stages II or IIIA, completely removed in surgery	Intracutaneously once weekly for 3 weeks beginning 2–7 weeks (no later than 49 days) after surgery, and then subcutaneously once monthly for 2 years, regardless of disease progression	Phase II, completed	NCT00006470
GSK 249553 vaccine	Recombinant MAGE-A3 protein combined with an immunostimulant	NSCLC, stages IB or II, complete surgical resection	Intramuscular injection; 5 doses at 3-week intervals and 8 doses at 3-month intervals	Phase II, completed	NCT00290355 [98]
GV 1001	Telomerase peptide vaccine	NSCLC, stages IIIA and IIIB received concurrent chemoradiotherapy	Intradermal	Phase II, completed	NCT00509457
DC vaccines	Tumor neoantigen primed dendritic cell vaccines	NSCLC has undergone a curative resection or ablation	Subcutaneous, at 2–3 week intervals for a total of 3–5 times	Phase I, recruiting	NCT04147078
V934/935	V934/V935 human telomerase reverse transcriptase (hTERT) vaccination	NSCLC stages I to III	Intramuscular	Phase I, completed	NCT00753415 [99]

Abbreviations: DC, dendritic cell; EGF, epidermal growth factor; NSCLC, non-small cell lung cancer.

## Data Availability

This review manuscript did not report any data.

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
