# Peer review of "Immunotherapy and Vaccination in Surgically Resectable Non-Small Cell Lung Cancer (NSCLC)"

_vaccines, 2021, doi:10.3390/vaccines9070689_

Round 1

Reviewer 1 Report

The manuscript by Chiu et. al. provides a comprehensive review of recent advances in the immunotherapy and vaccination in surgically resectable non-small cell lung cancer (NSCLC).   The manuscript reviewed available literature on immune checkpoint inhibitor therapies which are used alone or in combination with other therapies to treat NSCLC.  Both anti-CTLA-4 and anti-PD1 antibody therapies are covered well.  The mechanism of action of anti-PD-1/PD-L1 and anti-CTLA-4 antibodies in anti-cancer immunotherapies is depicted nicely in Figure 1.  Use of these antibodies in both post-operation adjuvant therapy and neoadjuvant therapy settings are discussed. Tables are used to list the past and ongoing clinical trials where immune checkpoint inhibitors were used alone or in combination with other approaches.  Overall, this is a very well written review article and detail oriented. 

Following are comments which needs to be addressed:

1) There is a condition called “hyper progression” in subset of patients during immune checkpoint inhibitor therapy which is not discussed. Please include relevant literature and discuss about. Following is a review on it:

Chen, Y., Hu, J., Bu, F. et al. Clinical characteristics of hyperprogressive disease in NSCLC after treatment with immune checkpoint inhibitor: a systematic review and meta-analysis. BMC Cancer 20, 707 (2020). https://doi.org/10.1186/s12885-020-07206-4

2) Line 20: “that the five-year survival rate is lower than 70% for IB” This needs to be modified to show exact number.  “lower than 70%” is misleading.

3) Line 71: Rephrase, “anti-PD1 Immune Checkpoint Inhibitors” to “anti-PD1 Immune Checkpoint antibodies “

4) Line 142: Rephrase, “anti-CTLA-4 Inhibitors” to “anti-CTLA-4 Inhibitors antibodies”

5) Lines 238 to 241: Refer antibodies as ‘antibody’ or ‘mAb’ not as IgG

6) Line 299: “Vaccination for Tumor Prevention in NSCLC” change to “ Therapeutic vaccination in NSCLC”

Author Response

Response to Reviewer 1 Comments

The manuscript by Chiu et. al. provides a comprehensive review of recent advances in the immunotherapy and vaccination in surgically resectable non-small cell lung cancer (NSCLC).  The manuscript reviewed available literature on immune checkpoint inhibitor therapies which are used alone or in combination with other therapies to treat NSCLC.  Both anti-CTLA-4 and anti-PD1 antibody therapies are covered well.  The mechanism of action of anti-PD-1/PD-L1 and anti-CTLA-4 antibodies in anti-cancer immunotherapies is depicted nicely in Figure 1.  Use of these antibodies in both post-operation adjuvant therapy and neoadjuvant therapy settings are discussed. Tables are used to list the past and ongoing clinical trials where immune checkpoint inhibitors were used alone or in combination with other approaches.  Overall, this is a very well written review article and detail oriented.

Point 1: There is a condition called “hyper progression” in subset of patients during immune checkpoint inhibitor therapy which is not discussed. Please include relevant literature and discuss about. Following is a review on it:

Chen, Y., Hu, J., Bu, F. et al. Clinical characteristics of hyperprogressive disease in NSCLC after treatment with immune checkpoint inhibitor: a systematic review and meta-analysis. BMC Cancer 20, 707 (2020). https://doi.org/10.1186/s12885-020-07206-4

Response 1: We added a paragraph to discuss about hyperprogressive disease (HPD) in NSCLC treated with ICIs and added references as suggested.

Previous studies reported hyperprogressive disease (HPD) in NSCLC treated with ICIs [Choi,Kim, Chen]. HPD was characterized by rapid tumor progression after the initiation of ICIs therapy, and was associated with poor prognosis [Choi ,Kim, Chen]. In several previous clinical analysis, the risk factors associated with HPD in NSCLC with ICIs treatment are serum lactate dehydrogenase (LDH) > upper limit of normal, neutrophil-to-lymphocyte ratio (NLR) of ≥3, liver metastasis and > 2 metastatic sites [Kim, Chen, Zhang]. In early-stage resectable NSCLC receiving neoadjuvant and/or adjuvant ICIs, the pretreatment LDH and NLR should be tested and imaging evaluation should be closely followed in patient with increased risk of HPD.

Point 2: Line 20: “that the five-year survival rate is lower than 70% for IB” This needs to be modified to show exact number. “lower than 70%” is misleading.

Response 2: We corrected this sentence as suggested.

The revised sentence is “In previous studies on the long-term follow-up of post-operative NSCLC, the results showed that the five-year survival rate was about 65% for stage IB and about 35% for stage IIIA diseases.”

Point 3: Line 71: Rephrase, “anti-PD1 Immune Checkpoint Inhibitors” to “anti-PD1 Immune Checkpoint antibodies “ 

Response 3: We corrected it in revised manuscript as suggested.

Point 4: Line 142: Rephrase, “anti-CTLA-4 Inhibitors” to “anti-CTLA-4 Inhibitors antibodies” 

Response 4: We corrected it in revised manuscript as suggested.

Point 5: Lines 238 to 241: Refer antibodies as ‘antibody’ or ‘mAb’ not as IgG

Response 5: We corrected it in revised manuscript as suggested.

Point 6: Line 299: “Vaccination for Tumor Prevention in NSCLC” change to “ Therapeutic vaccination in NSCLC”

Response 6: We corrected it in revised manuscript as suggested.

Reviewer 2 Report

see attached file

Author Response

Response to Reviewer 2 Comments

The paper gives a wide overview of the current use of anti-PD(L)-1 and anti-CTLA-4 antibodies in NSCLC, from the mechanisms of action, to the main clinical trials that led to their approval for advanced disease, to their application in the pre-operative or post-operative setting for early-stage disease. However, considering the title of the review, the latter point seems to be the minor part of the work. Thus, the

authors should consider to expand it, in order to be more consistent with the presumed aim of their review and to offer something more than what can be already found in literature. Some suggestions could be:

Point 1: To stress the rationale of neoadjuvant immunotherapy, citing other works that demonstrated its usefulness (e.g. Jia XH, Xu H, Geng LY, Jiao M, Wang WJ, Jiang LL, Guo H. Efficacy and safety of neoadjuvant immunotherapy in resectable nonsmall cell lung cancer: A meta-analysis. Lung Cancer. 2020 Sep;147:143-153) and are evaluating pre-clinical results (e.g. Bersanelli M, Tiseo M, Banna GL. Nivolumab plus Ipilimumab in Non-Small-Cell Lung Cancer. N Engl J Med. 2020;382(9):874-5; Reuss JE, Anagnostou V, Cottrell TR, Smith KN, Verde F, Zahurak M, et al. Neoadjuvant nivolumab plus ipilimumab in resectable non-small cell lung cancer. J Immunother Cancer. 2020;8(2));  

Response 1: We cited the recommended publications as suggested, and added discussion with these citations in revised manuscript.

A previous report found that nivolumab plus ipilimumab therapy had the trend of more effective in current or former smokers than never smokers base on the results of CheckMate 227 trial. Another clinical study showed that neoadjuvant nivolumab plus ipilimumab in resectable NSCLC is feasible, and all the patients enrolled in the study were active and former smokers. A previous meta-analysis review showed that neoadjuvant immunotherapy was more effective than neoadjuvant chemotherapy re-garding the MPR and pathological complete response (PCR) in resectable NSCLC. In the same analysis, the surgical resection rate was also similar between neoadjuvant immunotherapy and neoadjuvant chemotherapy (88.7% VS. 70-90%).

Point 2: To explain briefly the biologic rationale of the combination of chemotherapy and immunotherapy, as it is tested in the trials reported in the manuscript;

Response 2: We added a paragraph to discuss “the biologic rationale of the combination of chemotherapy and immunotherapy”. as suggested in revised manuscript. Additional references were also added to address this comment.

Cytotoxic chemotherapy augments the immunogenicity of cancer cells through induc-ing antigenicity and the adjuvanticity. The immunogenic cell death (ICD) is asso-ciated with adaptive stress response which promotes the maturation of dendritic cells (DCs). In a lung cancer mouse model, chemotherapy promotes ICD pathway to en-hance the anti-tumor ability of anti-PD-1 and anti-CTLA4 antibodies. In addi-tion, chemotherapy might have off-target effects on suppressing myeloid derived sup-pressor cells (MDSCs) or regulatory T (Treg) cells to stimulate anti-tumor immunity. Together, these indicated that chemotherapy in combination with ICIs success-fully improved survival of metastatic NSCLC patients.

Point 3: Considering the context of early-stage disease, a short paragraph regarding the combination of immunotherapy and radiotherapy in this setting could be appropriate, too;

Response 3: We added a paragraph and references to discuss about the combination of immunotherapy and radiotherapy as suggested.

There are remaining some early-stage NSCLC patients do not receive surgery because of the reasons including poor cardio-pulmonary reserve, extremely old age, poor performance status and personal refusal. Therefore, radiotherapy such as stereotactic ablative radiotherapy (SABR) can be an alternative treatment for early-stage NSCLC patients who are unable to receive surgery. Previous studies had shown that local radiation therapy can stimulate the release of tumor-associated antigens (TAAs) and damage-associated molecular patterns (DAMPs). The TAAs and DAMPs promote immune cell priming and destruct immunosuppressive tumor-supporting stroma, and these result in the enhancement of anti-cancer effect of ICIs in NSCLC. The efficacy of ICIs enhanced by radiotherapy is also called abscopal effect, and compatible with the promising results shown in PACIFIC trial. Using the combination of local radiation therapy and ICIs to improve local control and survival in early-stage NSCLC is warranted in future clinical trials.

Point 4: To increase the number of already available results of trials testing immunotherapy in the neoadjuvant setting (e.g B. Besse JA, N. Cozic, N. Chaput-Gras, D. Planchard, L. Mezquita, et al. Neoadjuvant atezolizumab (A) for resectable non-small cell lung cancer (NSCLC): Results from the phase II PRINCEPS trial. Ann Oncol. 2020;31(54):S794; Gao S, Li N, Gao S, Xue Q, Ying J, Wang S, et al. Neoadjuvant PD-1 inhibitor (Sintilimab) in NSCLC. J Thorac Oncol. 2020;15(5):816-26; etc.)

Response 4: We added the trials with available results testing immunotherapy in the neoadjuvant setting in table 2 as suggested.

Point 5: To cite and discuss some of the main ongoing trials in the same setting (e.g. Eichhorn F, Klotz LV, Bischoff H, Thomas M, Lasitschka F, Winter H, et al. Neoadjuvant anti-programmed Death-1 immunotherapy by Pembrolizumab in resectable nodal positive stage II/IIIa non-small-cell lung cancer (NSCLC): the NEOMUN trial. BMC Cancer. 2019;19(1):413; M. Wislez JM, A. Lavole, G. Zalcman, O. Carre, T. Egenod, et al. Neoadjuvant durvalumab in resectable non-small cell lung cancer (NSCLC): Preliminary results from a multicenter study (IFCT-1601 IONESCO). Ann Oncol. 2020;31(54):S794; Hong MH AB, Kim HR, Lim SM, Lee SY, Park SY, et al. Interim Analysis of Neoadjuvant Chemoradiotherapy (N-CRT) and Durvalumab for Potentially Resectable Stage III NSCLC. Presented at 2020 World Conference on Lung Cancer; January 30, 2021; Singapore, Abstract FP03.02; etc.). Another table summarizing these ongoing trials could give an useful update to the topic.

Response 5: We cited ongoing clinical trials using immunotherapy with or without chemotherapy as neoadjuvant therapy for resectable NSCLC patients as suggested. We also added additional table 3. to summarize these ongoing trials

At present, several ongoing clinical trials are investigating the use of ICIs with or without chemotherapy as neoadjuvant therapy in resectable NSCLC (Table 3.). Several previous early-phase (phase I & II) had shown that ICIs with or without chemotherapy were feasible and effective as neoadjuvant therapy before surgery. Therefore, four main phase III clinical trials (KEYNOTE 617, CheckMate 816, IMpower 030, AE-GEAN) are conducted and ongoing now. All the four trials enrolled control groups, and explore the consolidation ICIs therapy after surgery. These four clinical trials are expected to completed in 2024.

Point 6: The paragraph regarding ICIs as adjuvant therapy could be expanded, too, describing briefly the trials cited in text.

Response 6: We extended the discussion about the four ongoing adjuvant ICIs therapy clinical trials in revised manuscript as suggested.

Though the design of the four ongoing trials is similar, there is a little difference among the 4 ongoing trials. First, post-operative platinum-based chemotherapy before randomized to atezolizumab or best supportive care group is a required treatment for participants of the IMpower010 trial whether post-operative chemotherapy is optional for the participants of the other 3 ongoing trials. Second, the patients in control group of IMpower010 and ANVIL trials receive best supportive care or observation, and the patients in control group of the other 2 ongoing trials (PEARLS and BR31) receive placebo. Patients in the BR31 trial would have the tests EGFR mutation and ALK rearrangement for further sub-group analysis. Patients with EGFR mutation or ALK rearrangement would be excluded in the ANVIL trial. The tests of EGFR mutation and ALK rearrangement are not mandatory in PEARLS and IMpower010 trials. All the trials have the test of tumor tissue PD-L1 expressions for further subgroup analysis in the future. The results of the four ongoing trials will provide information on ICIs with or without chemotherapy as post-operative adjuvant therapy for clinical practice. 

Point 7: Moreover, the authors could consider to describe weaknesses and strengths of these trials, and give a comparison of advantages and disadvantages of neoadjuvant and adjuvant approaches.

Response 7: We added a section 3.3 to compare the neoadjuvant and adjuvant therapy as suggested.

According to the results of pre-clinical study by Cascone et al., the neoadjuvant ICIs seem to contribute better survival benefit than adjuvant setting has in mouse model. The complete clinical trials showed that ICIs with or without chemotherapy as neoadjuvant therapy achieved pathological response and contribute to complete surgical resection. However, there were remaining some NSCLC patients receiving neoadjuvant therapy did not receive surgical resection finally because of complication or disease progression in neoadjuvant therapy. In the main four ongoing trials investigating ICIs as adjuvant therapy for post-surgery NSCLC patients, the enrolled patients were required to have complete surgical resection (R0). However, some NSCLC patients have incomplete surgical resection in real-world clinical practice. Post-operative adjuvant therapy such as chemotherapy and radiotherapy are suggested for incomplete resection NSCLC patients. However, the survival benefit of post-operative conventional chemotherapy and radiotherapy is limited for incomplete resection NSCLC patients, and the prognosis of these patients are not well. ICIs in addition to chemotherapy or radiotherapy may provide survival benefit for incomplete resection NSCLC patients, but the 4 ongoing trials of adjuvant ICIs can not answer this clinical query.

In currently ongoing four main phase III clinical trials (KEYNOTE 617, CheckMate 816, IMpower 030, AE-GEAN) with neoadjuvant chemotherapy plus ICIs or placebo, post-operative consolidation ICIs therapy is administrated in the treatment group patients. These four clinical trials will provide clear evidences on the efficacy of ICIs administrated before and after surgery in early-stage and resectable NSCLC.

Point 8: The paragraph “vaccination for tumor prevention in NSCLC” should be better organized with sub-paragraphs, considering the fact that only the last part is really focused on this topic, and therefore should be expanded describing the main trials summarized in Table 4. The authors could consider to expand the text about the complexity and the dynamic interactions between immune cells and tumor microenviroment and about the possible strategies to elicit an immune response even in a suppressive environment. Furthermore, while in the abstract authors write that vaccination is being tested also in the adjuvant setting, any mention and discussion about it can be found in the text. Another interesting point to talk about could be the rationale of combining vaccines and immunotherapies.

Response 8: We re-organized with sub-paragraphs in the section of “Therapeutic vaccination in NSCLC” in and discussed the main trials summarized in Table 5. in revised manuscript as suggested.

    Clinical trials of preventive vaccines for lung cancer had been conducted. Early phase I trial was conducted to evaluate the effect of CIMAvax-EGF on the prevention of lung cancer development in high-risk patients, like family history of lung cancer or chronic pulmonary obstructive disease, or recurrence in patients with stage IB to IIIA NSCLC (Clinical Trial number: NCT04298606). Another phase I trial studies the side effects and how well peptide vaccine works in preventing lung cancer in high risk groups of current and former smokers (Clinical Trial number: NCT03300817). For clinical trials of vaccines in patients with surgically resectable NSCLC, a nonrandomized pilot study reported early clinical experience with vaccine 1650-G, an allogeneic cellular vaccine using granulocyte macrophage colony stimulating factor as an adjuvant. 1650-G is safe, biologically reliable and comparatively inexpensive, and could generate a robust and unequivocal immunological response. The relative frequency and kinetics of the response appears similar to that achieved with dendritic cell vaccines. Although therapeutic efficacy is unknown, immature dendritic cell vaccine preparation, pulsed with apoptotic tumor cells, has similar biologic efficacy to autologous dendritic cell vaccines matured with dendritic cell/T cell-derived maturation factor-matured preparation in NSCLC patients. The MAGE-A3 protein is expressed in approximately 35% of patients with resectable NSCLC patients. A double-blind, randomized, placebo-controlled phase II study investigated recombinant MAGE-A3 protein combined with an immunostimulant in completely resected MAGE-A3–positive stage IB to II NSCLC patients. Although there was no statistically significant improvement in disease-free interval and overall survival, postoperative MAGE-A3 immunization proved to be feasible with minimal toxicity. Human telomerase reverse transcriptase (hTERT) is an antigen that may represent a target for a novel anti-cancer strategy. A pilot, phase I study evaluated a prime-boost immunization regimen based on V935 (an adenoviral type 6 vector vaccine expressing a modified version of hTERT), administered alone or in combination with V934 (a DNA plasmid that expresses the same antigen), and the results revealed that the safety and feasibility of V934/V935 hTERT vaccination in cancer patients with solid tumors, including NSCLC.

  We expand the text in the revised manuscript as the reviewer’ suggestion about (1) the complexity and the dynamic interactions between immune cells and tumor microenvironment (2) the possible strategies to elicit an immune response in a suppressive environment.

The complex and dynamic nature of the interactions between immune cells and tumor microenvironment could influence tumor growth, invasion, and metastasis. The interactions consisting of cellular components including various myeloid and lymphoid cells, fibroblasts and endothelial cells that via direct interactions or biochemical cues (auto-, para-, and endocrine signaling) to communicate with tumor cells. Non-cellular components consisting of extracellular matrix, mechanical pressure and tumorigenic conditions like acidity, hypoglycemia and hypoxia that impact tumor behaviour. These components are essential to stimulate the heterogeneity of tumor cells, clonal evolution and increase the resistance leading to tumor progression and metastasis.

The fate of a tumor is dependent on dynamic properties of anti- to protumorigenic tumor microenvironment. The antitumorigenic tumor microenvironment contain normal fibroblasts, dendritic cells, natural killer (NK) cells, cytotoxic T cells, and M1-activated tumor-associated macrophages involving the activity of pro-inflammatory cytokines. The protumorigenic tumor microenvironment contain immunosuppressive effects of M2-activated tumor-associated macrophages involving production of anti-inflammatory cytokines, myeloid-derived suppressor cells, regulatory T cells and B cells, cancer-associated fibroblasts producing aberrant extracellular matrix, TIE2-expressing monocytes, and mast cells with angiogenesis stimulatory activity. Similar to tumor-associated macrophages, neutrophils and T helper cells can have both pro- and antitumorigenic activity depending on tumor and immune context.

Activation of the immune system to combat cancer was an appealing method for decades. However, the tumor microenvironment including immunosuppressive immune cells certainly contribute to hamper immunotherapy. Any therapy aiming to reduce immunosuppression must be given simultaneously to other agents that activate immune reactions or counteract other suppressive arms such as the myeloid cell compartment.

Immune activating therapies included different treatment modalities such as tumor-peptide-based vaccines, oncolytic viruses, agonistic antibodies and immune cell therapies based on T cells, NK cells, and dendritic cells. Options to reduce the action of immunosuppressive cells or molecules was evaluated. For example, antibodies that blocks chemokine receptors may prevent accumulation of suppressive myeloid cells such as M2 macrophages in the tumor microenvironment, the small-molecule mebendazole polarizes macrophages toward the anti-tumoral M1 phenotype and common chemotherapeutics such as gemcitabine, or tyrosine kinase inhibitors, reduces both MDSCs and/or regulatory T cells in patients. Such agents may be of high interest to combine with ICIs and activating immunotherapeutics. Agents affecting angiogenesis may be of interest since they limit blood supply and normalize the dysregulated blood vessels in the tumor tissues. Such a normalization may allow for better lymphocyte attachment, rolling, and transmigration into the tumor sites.

We apologized that the description “ICIs and vaccination as adjuvant therapies in post-operative NSCLC” in the abstract, which was not further mentioned and discussed in the main text. 

Cancer vaccines can elicit specific T cell responses, whereas immune checkpoint blockade boosts inactivated responses of effector T cells, vaccination can potentially activate naive T cells with tumor specificity and in this way broaden the tumor-specific immune responses. 

However, clinical trials of the combinations of ICIs and vaccination in post-operative NSCLC was scarce, only some trials in advanced NSCLC, and need further study in the future to conduct in these surgically resectable NSCLC patients. Therefore, we deleted this sentence in the abstract.   

 We add the rationale of combining vaccines and immunotherapies in the revised manuscript.

Some cancer patients does not benefit from immunotherapy, and this low response rate may be related to a limited specific T cell response developed against tumor cells, especially for tumors with a low mutational burden. The efficacy of immunotherapy appears to depend on preexisting intratumoral CD8+ T cells underlining the necessity to induce these cytotoxic T cells with vaccination. Cancer vaccines can generate tumor-specific T cells in periphery or in situ tumors and are able to drive activated peripheral T cells into the tumor microenvironment leading to increased tumor-infiltrating lymphocytes. Moreover, vaccine-mediated tumor cell death leads to the release of more cascade antigens and induces stronger immune responses specific to antigens (i.e., antigen cascade or epitope spreading).  Whereas ICIs boosts inactivated responses of effector T cells, vaccination can potentially activate naive T cells with tumor specificity and in this way broaden the tumor-specific immune responses. Therefore, combining a cancer vaccine, which can elicit specific T cell responses, with ICIs represents an attractive therapeutic option.

Point 9: The paragraph regarding future perspectives should be improved, discussing about other challenges in addition to the only oncogene-addicted disease: radiologic evaluation of response and the differences with the pathological response, surrogate endpoints of OS, timing and type of surgery, duration of treatment, predictive biomarkers. A summarizing figure in this regard could be eventually added.

Response 9: We had re-wrote and re-organized the section of future perspectives, and added figure 2. to address this comment.

5.1. Imaging Evaluation of Response before Surgery and Post-Operative Follow-up

The ICIs treatment has distinctive response patterns and adverse events related to biologic mechanism of anticancer activity, and the imaging evaluation of ICIs treatment is a challenge. Pseudoprogression is a phenomenon described that radiological image showed tumor progression with enlargement and/or appearance of new lesions initially, and followed by stabilization or regression of tumor without additional treatment. The causes of psuedoprogression may result from the ongoing tumor growth before achievement of immune response, the cytotoxic T lymphocytes recruitments, and inflammatory process surrounding tumor micro-environment. The incidence of pseudoprogression in NSCLC with ICIs treatment is about 5%. If pseudoprogression occurs in neoadjuvant therapy, it may affect the judgement and decision of surgery. Careful evaluation by using multiple images including radiological plain film, sonography, computed tomography (CT) scan, magnetic resonance imaging (MRI) and fluorodeoxyglucose (FDG)-positron emission tomography (PET) before surgery should be considered. In patients receiving post-operative adjuvant therapy with ICIs, pseudoprogression should be alert because it may mislead some patients in response to the therapy. Procedures including needle aspiration, core tissue biopsy and/or surgery on tumors or new lesions may be needed to determined pseudoprogression or true-progression in these patients.

5.2. Predictive Biomarkers in Therapy

      The issue of looking for biomarkers to predict the efficacy of immunotherapy in NSCLC emerges, but there is no idea predictive biomarker of immunotherapy for the complexity and diversity in tumor immune environment. PD-L1 expression and bTMB levels were reported to correlate with response rate and PFS of anti-PD-L1 ICIs in the treatment of advanced NSCLC. Pretreatment PD-L1 expression and TMB levels were also tested in neoadjuvant ICIs trials of early-stage NSCLC including NEOSTAR, LCMC3, NADIM and Forde et al. trials. The MPR was positively and significantly associated with the PD-L1 expression level in NEOSTAR trial, but was not significantly associated with the PD-L1 expression level in LCMC3, NADIM and Forde et al. studies. TMB was associated with MPR in the study of Forde et al., but was not observed in LCMC3 trial [78,79,89]. Heterogenous associations between PD-L1 expression, TMB levels and MPR were observed in these clinical trials. The PD-L1 expression and TMB levels should be carefully and interpreted before and during neoadjuvant therapy.

In adjuvant ICIs setting, the association between PD-L1 expression level and outcome will be analyzed and shown after the four ongoing trials completed in the future.

Previous studies reported hyperprogressive disease (HPD) in NSCLC treated with ICIs. HPD was characterized by rapid tumor progression after the initiation of ICIs therapy, and was associated with poor prognosis. In several previous clinical analysis, the risk factors associated with HPD in NSCLC with ICIs treatment are serum lactate dehydrogenase (LDH) > upper limit of normal, neutrophil-to-lymphocyte ratio (NLR) of ≥3, liver metastasis and > 2 metastatic sites [Kim, Chen, Zhang]. In early-stage resectable NSCLC receiving neoadjuvant and/or adjuvant ICIs, the pretreatment LDH and NLR should be tested and imaging evaluation should be closely followed in patient with increased risk of HPD.

Previous clinical studies have shown that early-stage NSCLC harboring EGFR mutations had an increased risk of recurrence and metastasis. A recent study has demonstrated that consolidation with durvalumab (anti-PD-L1 inhibitor) did not benefit the survival in stage III NSCLC patients with EGFR mutation receiving induction-concurrent chemoradiation therapy. Consolidation therapy with durvalumab also increased immune-related adverse events in post-chemoradiotherapy EGFR-mutated stage III NSCLC. To date, osimertinib is the recommended post-operative adjuvant therapy for stages IB–IIIA NSCLC with EGFR mutation, based on the promising results shown in the ADAURA trial. Alectinib is an effective and safe targeted therapy for advanced NSCLC patients with ALK rearrangement. The use of alectinib as a post-surgery adjuvant therapy in stage IB–IIIA ALK-mutated NSCLC patients is currently being explored in a phase III clinical trial (ALINA trial, NCT03456076). For, EGFR- and ALK- mutated early-stage NSCLC, surgery followed by adjuvant TKIs may be considered priorly to ICIs. The other oncogene-addicted NSCLC (ex. BRAF, MET, HER2, RET, K-RAS and NTRK mutations), neoadjuvant and adjuvant ICIs can be considered for resectable NSCLC unless the appearance of more effective and safer target therapy than ICIs.

5.3. The Primary Endpoints: Pathological Response and Survival; Timing and Type of surgery

MPR was the endpoint employed most frequently in early-phase trials with neoadjuvant ICIs. For early-stage NSCLC, the treatment goal should focus on disease cure and prolonging survival. Therefore, the event-free survival (EFS) and OS are employed as primary endpoint by the recent ongoing trials (KEYNOTE 617, CheckMate 816, IMpower 030, AEGEAN). Timing of surgery is also a concerning issue in using ICIs as neoadjuvant therapy. In the design of clinical trials with neoadjuvant ICIs, the neoadjuvant therapies were administrated for 2-4 cycles. These studies reported that the radiological response assessed by Response Evaluation Criteria in Solid Tumors (RECIST) did not correlate with pathological response [78-102]. For example, Forde et al. showed that with 90% patients with pathological response in the study had radiological stable disease before surgery. In NEOSTAR, 11% of study patients were found to had the phenomenon of pesudoprogression in mediastinal lymph node radiologically, and no cancer cells were found in these lymph nodes pathologically in surgical resection. Progressive disease in neoadjuvant therapy which lead to the abandonment of surgery was also reported in these trials. In LCMC3 trial, 10 study patients were found inoperable in exploration or experienced disease progression in neoadjuvant atezolizumab. Regarding the HPD of ICIs treatment, NSCLC patients with risk factors may be considered surgery directly followed by adjuvant therapy. Fortunately, no unexpected immune-related-adverse event (irAE) lead to the delay of surgery was reported in these trials with neoadjuvant ICIs in early-stage NSCLC. In an early trial of neoadjuvant therapy with the combination of nivolumab and ipilimumab in melanoma, 18 of 20 study patients (90%) experienced grade 3 and 4 irAE, and cessation of therapy was required in most patients (17 of 18).

Regarding the surgery type, extensive resection such as pneumonectomy in early-stage NSCLC contributed unfavorable outcome and severer morbidity compared with lobectomy. The pathological responses are achieved in part NSCLC patients receiving neoadjuvant ICIs, and surgery with more pulmonary volume may be considered and explored in future studies.

Cancer vaccination mainly explored in post-resected NSCLC for the prevention of recurrence. The comparison of efficacy between post-operative ICIs and vaccination is not clear. Vaccination may be an alternative choice for post-operative NSCLC patients who ever experience irAE in neoadjuvant therapy or at high risk of irAE.

The future strategy of neoadjuvant, surgery, and adjuvant therapies with immunotherapy is summarized in figure 2.

Point 10: In addition, regarding the paragraph “current immune checkpoint inhibitors in advanced NSCLC”:

- should be better illustrated the distinction between NSCLC "non oncogene" and "oncogene" addicted and how the unfavorable impact of the presence of genomic alterations in the latter one group is mitigated by the use of target therapies;.

Response 10: We had added a paragraph to discuss about the use and efficacy of of immune checkpoint inhibitors in oncogene-addicted and non-oncogene-addicted NSCLC as suggested. Additional references were also added in revised manuscript to addressed this comment.

A majority of NSCLC, especially adenocarcinomas harbor driver mutations and can be classified as oncogene-addicted NSCLC. Most of oncogene-addicted NSCLC had effective target therapies to their driver mutations including EGFR, anaplastic lymphoma kinase (ALK), ROS1, BRAF, MET, HER2, RET, K-RAS and NTRK. Several previous clinical studies have shown that anti-PD-1/PD-L1 ICIs had a significantly lower response rate and shorter survival in NSCLC patients harboring EGFR, ALK or ROS-1 mutations than non-oncogene-addicted NSCLC patients; therefore, EGFR-,ALK-,or ROS1- mutated NSCLC patients are generally not recruited in most clinical trials investigating the efficacy of first-line anti-PD-1/PD-L1 ICIs . To date, several TKIs targeting EGFR, ALK, and ROS1 mutations had shown promising efficacy in treating EGFR-,ALK-,or ROS1- mutated NSCLC patients (60%-80% response rate and 10-30 months of PFS). EGFR-TKIs (ex. gefitinib, erlotnib, afatinib, dacomitinib, and osimertinib), ALK inhibitors (ex. crizotinib, ceritinib, alectinib, brigatinib, and lorlatinib), and ROS1 inhibitor (ex. crizotinib) had been approved and wildly used in the treatment of EGFR-,ALK-,or ROS1- mutated NSCLC patients. Previous studies reported that the efficacy of anti-PD-L1 ICIs in NSCLC patients with BRAF, HER2, MET, KRAS or RET mutations was close to unselected NSCLC patients. For NSCLC with the rare driver mutations such as BRAF, HER2, MET, KRAS, or RET, ICIs are treatment choice for these patients before reliable target therapies available.

Point 11: should be specified that there are other genomic alterations in addiction to EGFR mutation and ALK rearrangement that are emerging (es. ROS1, HER2, MET, RET ecc) and for which specific treatments are being made available;

Response 11: As above comment, we added additional discussion and references about this point as suggested.

A majority of NSCLC, especially adenocarcinomas harbor driver mutations and can be classified as oncogene-addicted NSCLC. Most of oncogene-addicted NSCLC had effective target therapies to their driver mutations including EGFR, anaplastic lymphoma kinase (ALK), ROS1, BRAF, MET, HER2, RET, K-RAS and NTRK. Several previous clinical studies have shown that anti-PD-1/PD-L1 ICIs had a significantly lower response rate and shorter survival in NSCLC patients harboring EGFR, ALK or ROS-1 mutations than non-oncogene-addicted NSCLC patients; therefore, EGFR-,ALK-,or ROS1- mutated NSCLC patients are generally not recruited in most clinical trials investigating the efficacy of first-line anti-PD-1/PD-L1 ICIs . To date, several TKIs targeting EGFR, ALK, and ROS1 mutations had shown promising efficacy in treating EGFR-,ALK-,or ROS1- mutated NSCLC patients (60%-80% response rate and 10-30 months of PFS). EGFR-TKIs (ex. gefitinib, erlotnib, afatinib, dacomitinib, and osimertinib), ALK inhibitors (ex. crizotinib, ceritinib, alectinib, brigatinib, and lorlatinib), and ROS1 inhibitor (ex. crizotinib) had been approved and wildly used in the treatment of EGFR-,ALK-,or ROS1- mutated NSCLC patients. Previous studies reported that the efficacy of anti-PD-L1 ICIs in NSCLC patients with BRAF, HER2, MET, KRAS or RET mutations was close to unselected NSCLC patients. For NSCLC with the rare driver mutations such as BRAF, HER2, MET, KRAS, or RET, ICIs are treatment choice for these patients before reliable target therapies available.

Point 12: in the sub-paragraph 2.2 should be explained that the CASPIAN trial tested the combination of tremelimumab, durvalumab and chemotherapy in SLCL.

Response 12: We added the discussion about “CASPIAN trial tested the combination of tremelimumab, durvalumab and chemotherapy in SLCL” as suggested.

Durvalumab in combination with tremelimumab therapy had been explored in ear-ly-phase clinical trials, and these trials showed that this combination therapy had du-rable clinical activity and an acceptable safety profile in patients with pretreated and relapsed extensive-stage (ES)-SCLC patients. Therefore, tremelimumab plus durvalumab plus chemotherapy had been tested in a pivotal phase 3 clinical trial CAS-PIAN.

Point 13: Some mistakes can be found in the paper, such as in lines 139-140 where ‘Javelin Lung 200’ is repeated twice instead of ‘Javelin Lung 100’.

Response 13: We had corrected this mistake as suggested.

Reviewer 3 Report

This is a well-written review of immunotherapy for surgically resectable NSCLC. The review cites the relevant trials in a comprehensive manner. Regarding  section 4 on Vaccination for Tumor Prevention in NSCLC: the statement on line 311 is misleading.  Oncogenic driver mutations do not appear to be the source of the high value neoantigens in NSCLC. In line 346 the authors state that the "common initial management of pre-malignant lesions" is surgical removal.  This is not true for premalignancy in lung cancer and should be clarified.  The discussion in this section of premalignancy and preventive vaccines relates to other cancers but does not mention the extensive work underway in pulmonary premalignancy and how that may relate to lung cancer interception.

Author Response

Response to Reviewer 3 Comments

This is a well-written review of immunotherapy for surgically resectable NSCLC. The review cites the relevant trials in a comprehensive manner.

Point 1: Regarding section 4 on Vaccination for Tumor Prevention in NSCLC: the statement on line 311 is misleading. Oncogenic driver mutations do not appear to be the source of the high value neoantigens in NSCLC.

Response 1: Regarding the comment on “the misleading statement on line 311 due to the fact that oncogenic driver mutations do not appear to be the source of the high value neoantigens in NSCLC”, we deleted this statement in revised manuscript as suggested.

Point 2: In line 346 the authors state that the "common initial management of pre-malignant lesions" is surgical removal. This is not true for premalignancy in lung cancer and should be clarified. 

Response 2: Regarding the concern about “statement in line 346 that “common initial management of pre-malignant lesion” is surgical removal, and it is not true for lung cancer.” we deleted this statement in revised manuscript to avoid the confusion as suggested.

Point 3: The discussion in this section of premalignancy and preventive vaccines relates to other cancers but does not mention the extensive work underway in pulmonary premalignancy and how that may relate to lung cancer interception.

Response 3: We add a paragraph to discuss this point in the revised manuscript as suggested. We also discussed the main trials in NSCLC summarized in Table 5. including preventive vaccines in NSCLC to address this comment.

Clinical trials of preventive vaccines for lung cancer had been conducted. Early phase I trial was conducted to evaluate the effect of CIMAvax-EGF on the prevention of lung cancer development in high-risk patients, like family history of lung cancer or chronic pulmonary obstructive disease, or recurrence in patients with stage IB to IIIA NSCLC (Clinical Trial number: NCT04298606). Another phase I trial studies the side effects and how well peptide vaccine works in preventing lung cancer in high risk groups of current and former smokers (Clinical Trial number: NCT03300817). For clinical trials of vaccines in patients with surgically resectable NSCLC, a nonrandomized pilot study reported early clinical experience with vaccine 1650-G, an allogeneic cellular vaccine using granulocyte macrophage colony stimulating factor as an adjuvant. 1650-G is safe, biologically reliable and comparatively inexpensive, and could generate a robust and unequivocal immunological response. The relative frequency and kinetics of the response appears similar to that achieved with dendritic cell vaccines. Although therapeutic efficacy is unknown, immature dendritic cell vaccine preparation, pulsed with apoptotic tumor cells, has similar biologic efficacy to autologous dendritic cell vaccines matured with dendritic cell/T cell-derived maturation factor-matured preparation in NSCLC patients. The MAGE-A3 protein is expressed in approximately 35% of patients with resectable NSCLC patients. A double-blind, randomized, placebo-controlled phase II study investigated recombinant MAGE-A3 protein combined with an immunostimulant in completely resected MAGE-A3–positive stage IB to II NSCLC patients. Although there was no statistically significant improvement in disease-free interval and overall survival, postoperative MAGE-A3 immunization proved to be feasible with minimal toxicity. Human telomerase reverse transcriptase (hTERT) is an antigen that may represent a target for a novel anti-cancer strategy. A pilot, phase I study evaluated a prime-boost immunization regimen based on V935 (an adenoviral type 6 vector vaccine expressing a modified version of hTERT), administered alone or in combination with V934 (a DNA plasmid that expresses the same antigen), and the results revealed that the safety and feasibility of V934/V935 hTERT vaccination in cancer patients with solid tumors, including NSCLC.

Reviewer 4 Report

The authors presented an interesting review about recent treatments in NSCLC patients, focusing on immunotherapy and vaccination in surgically resectable cancer cases. In my opinion, this paper is well-organized and well-written, with a nice picture explaining ICIs' mechanism of action and informative tables. 

I only suggest some corrections.

Line 114: substitute "uresectable" for "unresectable".

Please, substitute "CTLA4" for "CTLA-4" in some parts of the text, for examples lines 166 and 189.

Line 209: replace "druvalumab" with "durvalumab".

Line 210: substitute "lead" for "leads".

Line 222: replace "programmed Death-Li and" with "programmed Death-Ligand"

Line 225: "Table 1" is repeated twice. 

Line 290: replace "competed" with "completed".

Lines 331-332: the sentence "...which the prolonged median survival by 4.1 months,..." is not clear, is there a mistake?

Line 362: the word "immune" is in bold type, please correct it. 

Author Response

Response to Reviewer 4 Comments

The authors presented an interesting review about recent treatments in NSCLC patients, focusing on immunotherapy and vaccination in surgically resectable cancer cases. In my opinion, this paper is well-organized and well-written, with a nice picture explaining ICIs' mechanism of action and informative tables.

Point 1: Line 114: substitute "uresectable" for "unresectable".

Response 1: We corrected this in revised manuscript as suggested.

Point 2:  Please, substitute "CTLA4" for "CTLA-4" in some parts of the text, for examples lines 166 and 189.

Response 2: We corrected this in revised manuscript as suggested.

Point 3: Line 209: replace "druvalumab" with "durvalumab".

Response 3: We corrected this in revised manuscript as suggested.

Point 4: Line 210: substitute "lead" for "leads".

Response 4: We corrected it as suggested.

Point 5: Line 222: replace "programmed Death-Li and" with "programmed Death-Ligand"

Response 5: We corrected it as suggested.

Point 6: Line 225: "Table 1" is repeated twice.

Response 6: We corrected it as suggested.

Point 7: Line 290: replace "competed" with "completed".

Response 7: We corrected it as suggested.

Point 8: Lines 331-332: the sentence "...which the prolonged median survival by 4.1 months,..." is not clear, is there a mistake?

Response 8: We corrected it as suggested, and it should be “which prolonged median survival by 4.1 months, compared with the results in those treated with placebo”

Point 9: Line 362: the word "immune" is in bold type, please correct it.

Response 9: We corrected it as suggested.
